# Independent Policy Gradient Methods
# for Competitive Reinforcement Learning

**Constantinos Daskalakis**
costis@csail.mit.edu

**Dylan J. Foster**
dylanf@mit.edu

**Noah Golowich**
nzg@mit.edu

Massachusetts Institute of Technology

## Abstract

We obtain global, non-asymptotic convergence guarantees for independent learning algorithms in competitive reinforcement learning settings with two agents (i.e., zero-sum stochastic games). We consider an episodic setting where in each episode, each player independently selects a policy and observes only *their own* actions and rewards, along with the state. We show that if both players run policy gradient methods in tandem, their policies will converge to a min-max equilibrium of the game, as long as their learning rates follow a two-timescale rule (which is necessary). To the best of our knowledge, this constitutes the first finite-sample convergence result for independent policy gradient methods in competitive RL; prior work has largely focused on centralized, coordinated procedures for equilibrium computation.

## 1 Introduction

Reinforcement learning (RL)—in which an agent must learn to maximize reward in an unknown dynamic environment—is an important frontier for artificial intelligence research, and has shown great promise in application domains ranging from robotics [34, 41, 39] to games such as Atari, Go, and Starcraft [52, 64, 69]. Many of the most exciting recent applications of RL are game-theoretic in nature, with multiple agents competing for shared resources or cooperating to solve a common task in stateful environments where agents' actions influence both the state and other agents' rewards [64, 57, 69]. Algorithms for such *multi-agent reinforcement learning (MARL)* settings must be capable of accounting for other learning agents in their environment, and must choose their actions in anticipation of the behavior of these agents. Developing efficient, reliable techniques for MARL is a crucial step toward building autonomous and robust learning agents.

While single-player (or, non-competitive RL has seen much recent theoretical activity, including development of efficient algorithms with provable, non-asymptotic guarantees [15, 4, 33, 22, 2], provable guarantees for MARL have been comparatively sparse. Existing algorithms for MARL can be classified into *centralized/coordinated* algorithms and *independent/decoupled* algorithms [75]. Centralized algorithms such as self-play assume the existence of a centralized controller that joinly optimizes with respect to all agents' policies. These algorithms are typically employed in settings where the number of players and the type of interaction (competitive, cooperative, etc.) are both known a-priori. On the other hand, in independent reinforcement learning, agents behave myopically and optimize their own policy while treating the environment as fixed. They observe only local information, such as their own actions, rewards, and the part of the state that is available to them. As such, independent learning algorithms are generally more versatile, as they can be applied even in uncertain environments where the type of interaction and number of other agents are not known to the individual learners.

Both centralized [64, 57, 69] and independent [47, 26] algorithms have enjoyed practical success across different domains. However, while centralized algorithms have experienced recent theoretical development, including provable finite-sample guarantees [71, 6, 73], theoretical guarantees for independent reinforcement learning have remained elusive. In fact, it is known that independent algorithms may fail to converge even in simple multi-agent tasks [14, 67, 13]: When agents update their policies independently, they induce distribution shift, which can break assumptions made by classical single-player algorithms. Understanding when these algorithms work, and how to stabilize their performance and tackle distribution shift, is recognized as a major challenge in multi-agent RL [47, 30].

In this paper, we focus on understanding the convergence properties of independent reinforcement learning with *policy gradient methods* [72, 66]. Policy gradient methods form the foundation for modern applications of multi-agent reinforcement learning, with state-of-the-art performance across many domains [61, 62]. Policy gradient methods are especially relevant for continuous reinforcement learning and control tasks, since they readily scale to large action spaces, and are often more stable than value-based methods, particularly with function approximation [35]. Independent reinforcement learning with policy gradient methods is poorly understood, and attaining global convergence results is considered an important open problem [75, Section 6].

We analyze the behavior of independent policy gradient methods in Shapley's *stochastic game* framework [63]. We focus on two-player zero-sum stochastic games with discrete state and action spaces, wherein players observe the entire joint state, take simultaneous actions, and observe rewards simultaneously, with one player trying to maximize the reward and the other trying to minimize it. To capture the challenge of independent learning, we assume that each player observes the state, reward, and *their own* action, but not the action chosen by the other player. We assume that the dynamics and reward distribution are unknown, so that players must optimize their policies using only realized trajectories consisting of the states, rewards, and actions. For this setting, we show that—while independent policy gradient methods may not converge in general—policy gradient methods following a *two-timescale rule* converge to a Nash equilibrium. We also show that moving beyond two-timescale rules by incorporating optimization techniques from matrix games such as optimism [19] or extragradient updates [37] is likely to require new analysis techniques.

At a technical level, our result is a special case of a more general theorem, which shows that (stochastic) two-timescale updates converge to Nash equilibria for a class of nonconvex minimax problems satisfying a certain two-sided gradient dominance property. Our results here expand the class of nonconvex minimax problems with provable algorithms beyond the scope of prior work [74], and may be of independent interest.

## 2  Preliminaries

We investigate the behavior of independent learning in two-player zero-sum stochastic games (or, Markov games), a simple competitive reinforcement learning setting [63, 44]. In these games, two players—a *min-player* and a *max-player*—repeatedly select actions simultaneously in a shared Markov decision process in order to minimize and maximize, respectively, a given objective function. Formally, a two-player zero-sum stochastic game is specified by a tuple $\mathcal{G} = (\mathcal{S}, \mathcal{A}, \mathcal{B}, P, R, \zeta, \rho)$:

- $\mathcal{S}$ is a finite *state space* of size $S = |\mathcal{S}|$.

- $\mathcal{A}$ and $\mathcal{B}$ are finite *action spaces* for the min- and max-players, of sizes $A = |\mathcal{A}|$ and $B = |\mathcal{B}|$.

- $P$ is the *transition probability function*, for which $P(s' \mid s, a, b)$ denotes the probability of transitioning to state $s'$ when the current state is $s$ and the players take actions $a$ and $b$. In general we will have $\zeta_{s,a,b} := 1 - \sum_{s' \in \mathcal{S}} P(s' \mid s, a, b) > 0$; this quantity represents the probability that $\mathcal{G}$ *stops* at state $s$ if actions $a, b$ are played.

- $R : \mathcal{S} \times \mathcal{A} \times \mathcal{B} \to [-1, 1]$ is the *reward function*; $R(s, a, b)$ gives the immediate reward when the players take actions $a, b$ in state $s$. The min-player seeks to minimize $R$ and the max-player seeks to maximize it.[1]

- $\zeta := \min_{s,a,b}\{\zeta_{s,a,b}\}$ is a lower bound on the probability that the game stops at any state $s$ and choices of actions $a, b$. We assume that $\zeta > 0$ throughout this paper.

- $\rho \in \Delta(\mathcal{S})$ is the *initial distribution* of the state at time $t = 0$.

At each time step $t \geq 0$, both players observe a state $s_t \in \mathcal{S}$, pick actions $a_t \in \mathcal{A}$ and $b_t \in \mathcal{B}$, receive reward $r_t := R(s_t, a_t, b_t)$, and transition to the next state $s_{t+1} \sim P(\cdot \mid s_t, a_t, b_t)$. With probability $\zeta_{s_t, a_t, b_t}$, the game stops at time $t$; since $\zeta > 0$, the game stops eventually with probability 1.

A pair of (randomized) policies $\pi_1 : \mathcal{S} \to \Delta(\mathcal{A})$, $\pi_2 : \mathcal{S} \to \Delta(\mathcal{B})$ induces a distribution $\Pr^{\pi_1, \pi_2}$ of trajectories $(s_t, a_t, b_t, r_t)_{0 \leq t \leq T}$, where $s_0 \sim \rho$, $a_t \sim \pi_1(\cdot \mid s_t), b_t \sim \pi_2(\cdot \mid s_t)$, $r_t = R(s_t, a_t, b_t)$, and $T$ is the last time step before the game stops (which is a random variable). The *value function* $V_s(\pi_1, \pi_2)$ gives the expected reward when $s_0 = s$ and the plays follow $\pi_1$ and $\pi_2$:

$$V_s(\pi_1, \pi_2) := \mathbb{E}_{\pi_1, \pi_2}\left[ \sum_{t=0}^{T} R(s_t, a_t, b_t) \mid s_0 = s \right],$$

where $\mathbb{E}_{\pi_1, \pi_2}[\cdot]$ denotes expectation under the trajectory distribution given induced by $\pi_1$ and $\pi_2$. We set $V_\rho(\pi_1, \pi_2) := \mathbb{E}_{s \sim \rho}[V_s(x, y)]$.

**Minimax value.** Shapley [63] showed that stochastic games satisfy a minimax theorem: For any game $\mathcal{G}$, there exists a Nash equilibrium $(\pi_1^\star, \pi_2^\star)$ such that

$$V_\rho(\pi_1^\star, \pi_2) \leq V_\rho(\pi_1^\star, \pi_2^\star) \leq V_\rho(\pi_1, \pi_2^\star), \quad \text{for all } \pi_1, \pi_2, \tag{1}$$

and in particular $V_\rho^\star := \min_{\pi_1} \max_{\pi_2} V_\rho(\pi_1, \pi_2) = \max_{\pi_2} \min_{\pi_1} V_\rho(\pi_1, \pi_2)$. Our goal in this setting is to develop algorithms to find $\varepsilon$-approximate Nash equilibria, i.e. to find $\pi_1$ such that

$$\max_{\pi_2} V_\rho(\pi_1, \pi_2) \leq V_\rho(\pi_1^\star, \pi_2^\star) + \varepsilon, \tag{2}$$

and likewise for the max-player.

**Visitation distributions.** For policies $\pi_1, \pi_2$ and an initial state $s_0$, define the *discounted state visitation distribution* $d_{s_0}^{\pi_1, \pi_2} \in \Delta(\mathcal{S})$ by

$$d_{s_0}^{\pi_1, \pi_2}(s) \propto \sum_{t \geq 0} \Pr^{\pi_1, \pi_2}(s_t = s | s_0),$$

where $\Pr^{\pi_1, \pi_2}(s_t = s | s_0)$ is the probability that the game has not stopped at time $t$ and the $t$th state is $s$, given that we start at $s_0$. We define $d_\rho^{\pi_1, \pi_2}(s) := \mathbb{E}_{s_0 \sim \rho}[d_{s_0}^{\pi_1, \pi_2}(s)]$.

**Additional notation.** For a vector $x \in \mathbb{R}^d$, we let $\|x\|$ denote the Euclidean norm. For a finite set $\mathcal{X}$, $\Delta(\mathcal{X})$ denotes the set of all distributions over $\mathcal{X}$. We adopt non-asymptotic big-oh notation: For functions $f, g : \mathcal{X} \to \mathbb{R}_+$, we write $f = \mathcal{O}(g)$ if there exists a universal constant $C > 0$ that does not depend on problem parameters, such that $f(x) \leq Cg(x)$ for all $x \in \mathcal{X}$.

## 3   Independent Learning

**Independent learning protocol.** We analyze independent reinforcement learning algorithms for stochastic games in an episodic setting in which both players repeatedly execute arbitrary policies for a fixed number of episodes with the goal of producing an (approximate) Nash equilibrium.

We formalize the notion of independent RL via the following protocol: At each episode $i$, the min-player proposes a policy $\pi_1^{(i)} : \mathcal{S} \to \Delta(\mathcal{A})$ and the max-player proposes a policy $\pi_2^{(i)} : \mathcal{S} \to \Delta(\mathcal{B})$ independently. These policies are executed in the game $\mathcal{G}$ to sample a trajectory. The min-player observes only its own trajectory $(s_1^{(i)}, a_1^{(i)}, r_1^{(i)}), \ldots, (s_T^{(i)}, a_T^{(i)}, r_T^{(i)})$, and the max-player likewise observes $(s_1^{(i)}, b_1^{(i)}, r_1^{(i)}), \ldots, (s_T^{(i)}, b_T^{(i)}, r_T^{(i)})$. Importantly, each player is oblivious to the actions selected by the other.

We call a pair of algorithms for the min- and max-players an *independent distributed protocol* if (1) the players only access the game $\mathcal{G}$ through the oracle model above (*independent oracle*), and (2) the players can only use private storage, and are limited to storing a constant number of past trajectories and parameter vectors (*limited private storage*). The restriction on limited private storage aims to rule out strategies that orchestrate the players' sequences of actions in order for them to both reconstruct a good approximation of entire game $\mathcal{G}$ in their memory, then solve for equilibria locally. We note that making this constraint precise is challenging, and that similar difficulties with formalizing it arise even for two-player matrix games, as discussed in Daskalakis et al. [18]. In any event, the policy

gradient methods analyzed in this paper satisfy these formal constraints *and* are independent in the intuitive sense, with the caveat that the players need a very small amount of a-priori coordination to decide which player operates at a faster timescale when executing two-timescale updates. Because of the necessity of two-timescale updates, our algorithm does not satisfy the requirement of *strong independence*, which we define to be the setting that disallows any coordination to break symmetry so as to agree on differing "roles" of the players (such as differing step-sizes or exploration probabilities). As discussed further in Section 5.1, we leave the question of developing provable guarantees for strongly independent algorithms of this type as an important open question.

**Our question: Convergence of independent policy gradient methods.** Policy gradient methods are widely used in practice [61, 62], and are appealing in their simplicity: Players adopt continuous policy parameterizations $x \mapsto \pi_x$, and $y \mapsto \pi_y$, where $x \in \mathcal{X} \subseteq \mathbb{R}^{d_1}$, $y \in \mathcal{Y} \subseteq \mathbb{R}^{d_2}$ are parameter vectors. Each player simply treats $V_\rho(x, y) := V_\rho(\pi_x, \pi_y)$ as a continuous optimization objective, and updates their policy using an iterative method for stochastic optimization, using trajectories to form stochastic gradients for $V_\rho$.

For example, if both players use the ubiquitous REINFORCE gradient estimator [72], and update their policies with stochastic gradient descent, the updates for episode $i$ take the form[2]

$$x^{(i+1)} \leftarrow \mathcal{P}_{\mathcal{X}}(x^{(i)} - \eta_x \widehat{\nabla}_x^{(i)}), \quad \text{and} \quad y^{(i+1)} \leftarrow \mathcal{P}_{\mathcal{Y}}(y^{(i)} + \eta_y \widehat{\nabla}_y^{(i)}), \tag{3}$$

with

$$\widehat{\nabla}_x^{(i)} := R_T^{(i)} \sum_{t=0}^{T} \nabla \log \pi_x(a_t^{(i)} \mid s_t^{(i)}), \quad \text{and} \quad \widehat{\nabla}_y^{(i)} := R_T^{(i)} \sum_{t=0}^{T} \nabla \log \pi_y(b_t^{(i)} \mid s_t^{(i)}), \tag{4}$$

where $R_T^{(i)} := \sum_{t=0}^{T} r_t^{(i)}$, and where $x^{(0)}, y^{(0)}$ are initialized arbitrarily. This protocol is independent, since each player forms their respective policy gradient using only the data from their own trajectory. This leads to our central question:

*When do independent agents following policy gradient updates in a zero-sum stochastic game converge to a Nash equilibrium?*

We focus on an $\varepsilon$-greedy variant of the so-called *direct parameterization* where $\mathcal{X} = \Delta(\mathcal{A})^{|\mathcal{S}|}$, $\mathcal{Y} = \Delta(\mathcal{B})^{|\mathcal{S}|}$, $\pi_x(a \mid s) = (1 - \varepsilon_x)x_{s,a} + \varepsilon_x/|\mathcal{A}|$, and $\pi_y(a \mid s) = (1 - \varepsilon_y)y_{s,b} + \varepsilon_y/|\mathcal{B}|$, where $\varepsilon_x$ and $\varepsilon_y$ are exploration parameters. This is a simple model, but we believe it captures the essential difficulty of the independent learning problem.

**Challenges of independent learning.** Independent learning is challenging even for *simple stochastic games*, which are a special type of stochastic game in which only a single player can choose an action in each state, and where there are no rewards except in certain "sink" states. Here, a seminal result of Condon [14], establishes that even with oracle access to the game $\mathcal{G}$ (e.g., exact $Q$-functions given the opponent's policy), many naive approaches to independent learning can cycle and fail to approach equilibria, including protocols where (1) both players perform policy iteration independently, and (2) both players compute best responses at each episode. On the positive side, Condon [14] also shows that if one player performs policy iteration independently while the other computes a best response at each episode, the resulting algorithm converges, which parallels our findings.

Stochastic games also generalize two-player zero-sum matrix games. Here, even with exact gradient access, it is well-known that if players update their strategies independently using online gradient descent/ascent (GDA) with the same learning rate, the resulting dynamics may cycle, leading to poor guarantees unless the entire iterate sequence is averaged [19, 50]. To make matters worse, when one moves beyond the convex-concave setting, such iterate averaging techniques may fail altogether, as their analysis critically exploits convexity/concavity of the loss function. To give stronger guarantees—either for the last-iterate or for "most" elements of the iterate sequence— more sophisticated techniques based on two-timescale updates or negative momentum are required. However, existing results here rely on the machinery of convex optimization, and stochastic games— even with direct parameterization—are nonconvex-nonconcave, leading to difficulties if one attempts to apply these techniques out of the box.

In light of these challenges, it suffices to say that we are aware of no global convergence results for independent policy gradient methods (or any other independent distributed protocol, for that matter) in general finite state/action zero-sum stochastic games.

## 4   Main Result

We show that independent policy gradient algorithms following the updates in (3) converge to a Nash equilibrium, so long as their learning rates follow a *two-timescale* rule. The two-timescale rule is a simple modification of the usual gradient-descent-ascent scheme for minimax optimization in which the min-player uses a much smaller stepsize than the max-player (i.e., $\eta_x \ll \eta_y$), and hence works on a slower timescale (or vice-versa). Two-timescale rules help to avoid limit cycles in simple minimax optimization settings [31, 43], and our result shows that their benefits extend to MARL as well.

**Assumptions.**   Before stating the result, we first introduce some technical conditions that quantify the rate of convergence. First, it is well-known that policy gradient methods can systematically under-explore hard-to-reach states. Our convergence rates depend on an appropriately-defined *distribution mismatch coefficient* which bounds the difficulty of reaching such states, generalizing results for the single-agent setting [2]. While methods based on sophisticated exploration (e.g., [15, 33]) can avoid dependence on mismatch parameters, our goal here—similar to prior work in this direction [2, 8]—is to understand the behavior of standard methods used in practice, so we take the dependence on such parameters as a given.

Given a stochastic game $\mathcal{G}$, we define the *minimax mismatch* coefficient for $\mathcal{G}$ by:

$$C_{\mathcal{G}} := \max\left\{ \max_{\pi_2} \min_{\pi_1 \in \Pi_1^\star(\pi_2)} \left\| \frac{d_\rho^{\pi_1,\pi_2}}{\rho} \right\|_\infty, \max_{\pi_1} \min_{\pi_2 \in \Pi_2^\star(\pi_1)} \left\| \frac{d_\rho^{\pi_1,\pi_2}}{\rho} \right\|_\infty \right\}, \qquad (5)$$

where $\Pi_1^\star(\pi_2)$ and $\Pi_2^\star(\pi_1)$ each denotes the set of best responses for the min- (resp. max-) player when the max- (resp. min-) player plays $\pi_2$ (resp. $\pi_1$).

Compared to results for the single-agent setting, which typically scale with $\|d_\rho^{\pi^\star}/\rho\|_\infty$, where $\pi^\star$ is an optimal policy [2], the minimax mismatch coefficient measures the worst-case ratio for each player, given that their adversary best-responds. While the minimax mismatch coefficient in general is larger than its single-agent counterpart, it is still weaker than other notions of mismatch such as concentrability [54, 12, 25], which—when specialized to the two-agent setting—require that the ratio is bounded for *all* pairs of policies. The following proposition makes this observation precise.

**Proposition 1.**   There exists a stochastic game with five states and initial distribution $\rho$ such that $C_{\mathcal{G}}$ is bounded, but the concentrability coefficient $\max_{\pi_1,\pi_2} \left\| \frac{d_\rho^{\pi_1,\pi_2}}{\rho} \right\|_\infty$ is infinite.

Next, to ensure the variance of the REINFORCE estimator stays bounded, we require that both players use $\varepsilon$-greedy exploration in conjunction with the basic policy gradient updates (3).

**Assumption 1.**   Both players follow the direct parameterization with $\varepsilon$-greedy exploration: Policies are parameterized as $\pi_x(a \mid s) = (1 - \varepsilon_x)x_{s,a} + \varepsilon_x/|\mathcal{A}|$ and $\pi_y(a \mid s) = (1 - \varepsilon_y)y_{s,b} + \varepsilon_y/|\mathcal{B}|$, where $\varepsilon_x, \varepsilon_y \in [0, 1]$ are the *exploration parameters*.

We can now state our main result.

**Theorem 1.**   *Let $\epsilon > 0$ be given. Suppose both players follow the independent policy gradient scheme (3) with the parameterization in Assumption 1. If the learning rates satisfy $\eta_x \asymp \epsilon^{10.5}$ and $\eta_y \asymp \epsilon^6$ and the exploration parameters satisfy $\varepsilon_x \asymp \epsilon, \varepsilon_y \asymp \epsilon^2$, we are guaranteed that*

$$\mathbb{E}\left[ \frac{1}{N} \sum_{i=1}^N \max_{\pi_2} V_\rho(\pi_{x^{(i)}}, \pi_2) \right] - \min_{\pi_1} \max_{\pi_2} V_\rho(\pi_1, \pi_2) \leq \epsilon \qquad (6)$$

*after $N \leq \mathrm{poly}(\epsilon^{-1}, C_{\mathcal{G}}, S, A, B, \zeta^{-1})$ episodes.*

This represents, to our knowledge, the first finite-sample, global convergence guarantee for independent policy gradient updates in stochastic games. Some key features are as follows:

• Since the learning agents only use their own trajectories to make decisions, and only store a single parameter vector in memory, the protocol is independent in the sense of Section 3. However, an

important caveat is that since the players use different learning rates, the protocol only succeeds if this is agreed upon in advance.

• The two-timescale update rule may be thought of as a softened "gradient descent vs. best response" scheme in which the min-player updates their strategy using policy gradient and the max-player updates their policy with a best response to the min-player (since $\eta_x \ll \eta_y$). This is why the guarantee is asymmetric, in that it only guarantees that the iterates of the min-player are approximate Nash equilibria.[3] We remark that the gradient descent vs. exact best response has recently been analyzed for linear-quadratic games [76], and it is possible to use the machinery of our proofs to show that it succeeds in our setting of stochastic games as well.

• Eq. (13) shows that the iterates of the min-player have low error on average, in the sense that the expected error is smaller than $\epsilon$ if we select an iterate from the sequence uniformly at random. Such a guarantee goes beyond what is achieved by GDA with equal learning rates: Even for zero-sum matrix games, the iterates of GDA can reach limit cycles that remain a constant distance from the equilibrium, so that any individual iterate in the sequence will have high error [50]. While averaging the iterates takes care of this issue for matrix games, this technique relies critically on convexity, which is not present in our policy gradient setting. While our guarantees are stronger than GDA, we believe that giving guarantees that hold for individual (in particular, last) iterates rather than on average over iterates is an important open problem, and we discuss this further in Section 5.1.

• We have not attempted to optimize the dependence on $\epsilon^{-1}$ or other parameters, and this can almost certainly be improved.

The full proof of Theorem 1—as well as explicit dependence on problem parameters—is deferred to Appendix B. In the remainder of this section we sketch the key techniques.

**Overview of techniques.** Our result builds on recent advances that prove that policy gradient methods converge in single-agent reinforcement learning ([2]; see also [8]). These results show that while the reward function $V_\rho(\pi_x) = \mathbb{E}_{\pi_x}[\sum_{t=1}^T r_t \mid s_0 \sim \rho]$ is not convex—even for the direct parameterization—it satisfies a favorable *gradient domination* condition whenever a distribution mismatch coefficient is bounded. This allows one to apply standard results for finding first-order stationary points in smooth nonconvex optimization out of the box to derive convergence guarantees. We show that two-player zero-sum stochastic games satisfy an analogous *two-sided gradient dominance condition*.

**Lemma 1.** Suppose that players follow the $\varepsilon$-greedy direct parameterization of Assumption 1 with parameters $\varepsilon_x$ and $\varepsilon_y$. Then for all $x \in \Delta(\mathcal{A})^{|\mathcal{S}|}$, $y \in \Delta(\mathcal{B})^{|\mathcal{S}|}$ we have

$$V_\rho(\pi_x, \pi_y) - \min_{\pi_1} V_\rho(\pi_1, \pi_y) \leq \min_{\pi_1 \in \Pi_1^*(\pi_y)} \left\| \frac{d_\rho^{\pi_1, \pi_y}}{\rho} \right\|_\infty \left( \frac{1}{\zeta} \max_{\bar{x} \in \Delta(\mathcal{A})^{|\mathcal{S}|}} \langle \nabla_x V_\rho(\pi_x, \pi_y), x - \bar{x} \rangle + \frac{2\varepsilon_x}{\zeta^3} \right),$$

(7)

and an analogous upper bound holds for $\max_{\pi_2} V_\rho(\pi_y, \pi_2) - V_\rho(\pi_x, \pi_y)$.

Informally, the gradient dominance condition posits that for either player to have low regret relative to the best response to the opponent's policy, it suffices to find a near-stationary point. In particular, while the function $x \mapsto V_\rho(x, y)$ is nonconvex, the condition (7) implies that if the max-player fixes their strategy, all local minima are global for the min-player.

Unfortunately, compared to the single-agent setting, we are aware of no existing black-box minimax optimization results that can exploit this condition to achieve even asymptotic convergence guarantees. To derive our main results, we develop a new proof that two-timescale updates find Nash equilibria for generic minimax problems that satisfy the two-sided GD condition.

**Theorem 2.** *Let $\mathcal{X}$ and $\mathcal{Y}$ be convex sets with diameters $D_\mathcal{X}$ and $D_\mathcal{Y}$. Let $f : \mathcal{X} \times \mathcal{Y} \to \mathbb{R}$ be any, $\ell$-smooth, $L$-Lipschitz function for which there exist constants $\mu_x, \mu_y, \varepsilon_x,$ and $\varepsilon_y$ such that for all*

$x \in \mathcal{X}$ and $y \in \mathcal{Y}$,

$$\max_{\bar{x} \in \mathcal{X}, \|\bar{x} - x\| \leq 1} \langle x - \bar{x}, \nabla_x f(x, y) \rangle \geq \mu_x \cdot \left( f(x, y) - \min_{x' \in \mathcal{X}} f(x', y) \right) - \varepsilon_x, \tag{8}$$

$$\max_{\bar{y} \in \mathcal{Y}: \|\bar{y} - y\| \leq 1} \langle \bar{y} - y, \nabla_y f(x, y) \rangle \geq \mu_y \cdot \left( \max_{y' \in \mathcal{Y}} f(x, y') - f(x, y) \right) - \varepsilon_y. \tag{9}$$

*Then, given stochastic gradient oracles with variance at most $\sigma^2$, two-timescale stochastic gradient descent-ascent (Eq. (25) in Appendix C) with learning rates $\eta_x \asymp \epsilon^8$ and $\eta_y \asymp \epsilon^4$ ensures that*

$$\mathbb{E} \left[ \frac{1}{N} \sum_{i=1}^{N} \max_{y \in \mathcal{Y}} f(x^{(i)}, y) \right] - \min_{x \in \mathcal{X}} \max_{y \in \mathcal{Y}} f(x, y) \leq \epsilon \tag{10}$$

*within $N \leq \text{poly}(\epsilon^{-1}, D_\mathcal{X}, D_\mathcal{Y}, L, \ell, \mu_x^{-1}, \mu_y^{-1}, \sigma^2)$ episodes.*

A formal statement and proof of Theorem 2 are given in Appendix C. To deduce Theorem 1 from this result, we simply trade off the bias due to exploration with the variance of the REINFORCE estimator.

Our analysis of the two-timescale update rule builds on [43], who analyzed it for minimax problems $f(x, y)$ where $f$ is nonconvex with respect to $x$ but *concave* with respect to $y$. Compared to this setting, our nonconvex-nonconcave setup poses additional difficulties. At a high level, our approach is as follows. First, thanks to the gradient dominance condition for the $x$-player, to find an $\epsilon$-suboptimal solution it suffices to ensure that the gradient of $\Phi(x) := \max_{y \in \mathcal{Y}} f(x, y)$ is small. However, since $\Phi$ may not differentiable, we instead aim to minimize $\|\nabla \Phi_\lambda(x)\|_2$, where $\Phi_\lambda$ denotes the Moreau envelope of $\Phi$ (Appendix C.2). If the $y$-player performed a best response at each iteration, a standard analysis of nonconvex stochastic subgradient descent [20], would ensure that $\|\nabla \Phi_\lambda(x^{(i)})\|_2$ converges at an $\epsilon^{-4}$ rate. The crux of our analysis is to argue that, since the $x$ player operates at a much slower timescale than the $y$-player, the $y$-player approximates a best response in terms of function value. Compared to [43], which establishes this property using convexity for the $y$-player, we use the gradient dominance condition to bound the $y$-player's immediate suboptimality in terms of the norm of the gradient of the function $\psi_{t,\lambda}(y) := -(-f(x_t, \cdot))_\lambda(y)$, then show that this quantity is small on average using a potential-based argument.

## 5 Discussion

### 5.1 Toward Last-Iterate Convergence for Stochastic Games

An important problem left open by our work is to develop independent policy gradient-type updates that enjoy *last iterate convergence*. This property is most cleanly stated in the noiseless setting, with exact access to gradients: For fixed, constant learning rates $\eta_x = \eta_y = \eta$, we would like that if both learners independently run the algorithm, their iterates satisfy

$$\lim_{i \to \infty} x^{(i)} \to x^\star, \quad \text{and} \quad \lim_{i \to \infty} y^{(i)} \to y^\star.$$

Algorithms with this property have enjoyed intense recent interest for continuous, zero-sum games [19, 16, 50, 17, 40, 28, 53, 36, 27, 1, 5, 29]. These include Korpelevich's extragradient method [37], Optimistic Mirror Descent (e.g., [19]), and variants. For a generic minimax problem $f(x, y)$, the updates for the extragradient method take the form

$$x^{(i+1)} \leftarrow \mathcal{P}_\mathcal{X}(x^{(i)} - \eta \nabla_x f(x^{(i+1/2)}, y^{(i+1/2)})), \quad \text{and} \quad y^{(i+1)} \leftarrow \mathcal{P}_\mathcal{Y}(y^{(i)} + \eta \nabla_y f(x^{(i+1/2)}, y^{(i+1/2)})),$$

$$\text{where} \quad x^{(i+1/2)} \leftarrow \mathcal{P}_\mathcal{X}(x^{(i)} - \eta \nabla_x f(x^{(i)}, y^{(i)})), \quad \text{and} \quad y^{(i+1/2)} \leftarrow \mathcal{P}_\mathcal{Y}(y^{(i)} + \eta \nabla_y f(x^{(i)}, y^{(i)})).$$
$$\text{(EG)}$$

In the remainder of this section we show that while the extragradient method appears to succeed in simple two-player zero-sum stochastic games experimentally, establishing last-iterate convergence formally likely requires new tools. We conclude with an open problem.

As a running example, we consider von Neumann's *ratio* game [70], a very simple stochastic game given by

$$V(x, y) = \frac{\langle x, Ry \rangle}{\langle x, Sy \rangle}, \tag{11}$$

where $x \in \Delta(\mathcal{A})$, $y \in \Delta(\mathcal{B})$, $R \in \mathbb{R}^{A \times B}$, and $S \in \mathbb{R}_+^{A \times B}$, with $\langle x, Sy \rangle \geq \zeta$ for all $x \in \Delta(\mathcal{A})$, $y \in \Delta(\mathcal{B})$. The expression (11) can be interpreted as the value $V(\pi_x, \pi_y)$ for a stochastic game with a single

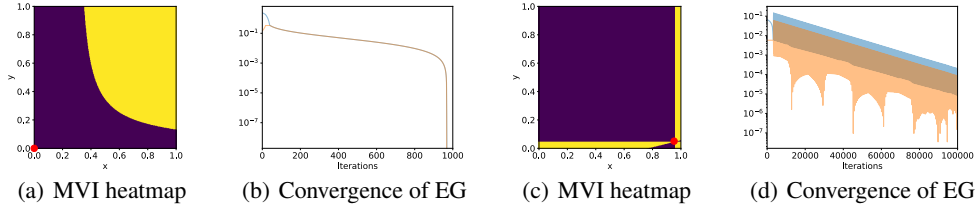

| (a) MVI heatmap | (b) Convergence of EG | (c) MVI heatmap | (d) Convergence of EG |

Figure 1: Figures (a) and (b) display plots for one ratio game, and Figures (c) and (d) display plots for another; the games' matrices are specified in Appendix D.1. Figures (a) and (c) plot the quantity $\text{sign}(\langle F(z), z - z^\star \rangle)$ for $z \in \Delta^2 \times \Delta^2$, parameterized as $z := (x, 1 - x, y, 1 - y)$; yellow denotes negative and purple denotes positive. The red dot denotes the equilibrium $z^\star$. Figures (b) and (d) plot convergence of extragradient with learning rate 0.01, initialized at $z_0 := (1, 0, 1, 0)$; note that $z_0$ is inside the region in which the MVI does not hold for each problem. The blue line plots the primal-dual gap $\max_{y'} V(x^{(i)}, y') - \min_{x'} V(x', y^{(i)})$ and the orange line plots the primal gap $\max_{y'} V(x^{(i)}, y') - V(x^\star, y^\star)$.

state, where the immediate reward for selecting actions $(a, b)$ is $R_{a,b}$, the probability of stopping in each round is $S_{a,b}$, and both players use the direct parameterization.[4] Even for this simple game, with exact gradients, we know of no algorithms with last iterate guarantees.

**On the MVI condition.** For nonconvex-nonconcave minimax problems, the only general tool we are aware of for establishing last-iterate convergence for the extragradient method and its relatives is the *Minty Variational Inequality* (MVI) property [24, 42, 49, 51, 27]. For $z = (x, y)$ and $F(z) := (\nabla_x f(x, y), -\nabla_y f(x, y))$, the MVI property requires that there exists a point $z^\star \in \mathcal{Z} := \mathcal{X} \times \mathcal{Y}$ such that

$$\langle F(z), z - z^\star \rangle \geq 0 \quad \forall z \in \mathcal{Z}. \tag{MVI}$$

For general minimax problems, the MVI property is typically applied with $z^\star$ as a Nash equilibrium [51]. We show that this condition fails in stochastic games, even for the simple ratio game in (12)

**Proposition 2.** Fix $\epsilon, s \in (0, 1)$ with $\epsilon < \frac{1-s}{2s}$. Suppose we take

$$R = \begin{pmatrix} -1 & \epsilon \\ -\epsilon & 0 \end{pmatrix}, \quad \text{and} \quad S = \begin{pmatrix} s & s \\ 1 & 1 \end{pmatrix}. \tag{12}$$

Then the ratio game defined by (12) has the following properties: (1) there is a unique Nash equilibrium $z^\star = (z^\star, y^\star)$ given by $x^\star = y^\star = (0, 1)$, (2) $\zeta \geq s$, (3) there exists $z = (x, y) \in \Delta(\mathcal{A}) \times \Delta(\mathcal{B})$ so that $\langle F(z), z - z^\star \rangle < 0$.[5]

Figure 1(a) plots the sign of $\langle F(z), z - z^\star \rangle$ for the game in (12) as a function of the players' parameters, which changes based on whether they belong to one of two regions, and Figure 1(b) shows that extragradient readily converges to $z^\star$ in spite of the failure of MVI. While this example satisfies the MVI property locally around $z^\star$, Figure 1(c) shows a randomly generated game (Appendix D.1) for which the MVI property fails to hold even locally. Nonetheless, Figure 1(d) shows that extragradient converges for this example, albeit more slowly, and with oscillations. This leads to our open problem.

**Open Problem 1.** *Does the extragradient method with constant learning rate have last-iterate convergence for the ratio game (11) for any fixed $\zeta > 0$?*

Additional experiments with *multi-state* games generated at random suggest that the extragradient method has last-iterate convergence for general stochastic games with a positive stopping probability. Proving such a convergence result for extragradient or for relatives such as the optimistic gradient method would be of interest not only because it would guarantee last-iterate convergence, but because it would provide an algorithm that is *strongly independent* in the sense that two-timescale updates are not required.

## 5.2 Related Work

While we have already discussed related work most closely related to our results, we refer the reader to Appendix A for a more extensive survey, both from the MARL and minimax perspective.

## 5.3 Future Directions

We presented the first independent policy gradient algorithms for competitive reinforcement learning in zero-sum stochastic games. We hope our results will serve as a starting point for developing a more complete theory for independent reinforcement learning in competitive RL and multi-agent reinforcement learning. Beyond Open Problem 1, there are a number of questions raised by our work.

Efroni et al. [23] have recently shown how to improve the convergence rates for policy gradient algorithms in the single-agent setting by incorporating optimism. Finding a way to use similar techniques in the multi-agent setting under the independent learning requirement could be another promising direction for future work.

Many games of interest are not zero-sum, and may involve more than two players or be cooperative in nature. It would be useful to extend our results to these settings, albeit likely for weaker solution concepts, and to derive a tighter understanding of the optimization geometry for these settings.

On the technical side, there are a number of immediate technical extensions of our results which may be useful to pursue, including (1) extending to linear function approximation, (2) extending to other policy parameterizations such as soft-max, and (3) actor-critic and natural policy gradient-based variants [2].

## Broader Impact

This is a theoretical paper, and we expect that the immediate ethical and societal consequences of our results will be limited. However, we believe that reinforcement learning more broadly will have significant impact on society. There is much potential for benefits to humanity in application domains including medicine and personalized education. There is also much potential for harm—for example, while reinforcement learning has great promise for self-driving cars and robotic systems, deploying methods that are not safe and reliable in these areas could lead to serious societal and economic consequences. We hope that research into the foundations of reinforcement learning will lead to development of algorithms with better safety and reliability.

## Acknowledgments and Disclosure of Funding

C.D. is supported by NSF Awards IIS-1741137, CCF-1617730 and CCF-1901292, by a Simons Investigator Award, and by the DOE PhILMs project (No. DE-AC05-76RL01830). D.F. acknowledges the support of NSF TRIPODS grant #1740751. N.G. is supported by a Fannie & John Hertz Foundation Fellowship and an NSF Graduate Fellowship.

## Footnotes

[1]We consider deterministic rewards for simplicity, but our results immediately extend to stochastic rewards.

[2]For a convex set $\mathcal{X}$, $\mathcal{P}_{\mathcal{X}}$ denotes euclidean projection onto the set.

[3]From an *optimization perspective*, the oracle complexity of finding a solution so that the iterates of both the min- and max-players are approximate equilibria is only twice as large as that in Theorem 1, since we may apply Theorem 1 with the roles switched.

[4] Since there is a single state, we drop the dependence on the initial state distribution.

[5] In fact, for this example the MVI property fails for all choices of $z^\star$, not just the Nash equilibrium.

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
