[Supplementary Material]

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

# A  Related work

Issues of independence in MARL have enjoyed extensive investigation. We refer the reader to Zhang et al. [75] for a comprehensive overview and discuss some particularly relevant related work below.

**Stochastic games.**  Beginning with their introduction by Shapley [63], there is a long line of work developing computationally efficient algorithms for multi-agent RL in stochastic games [44, 32, 11]. While centralized, coordinated MARL algorithms such as self-play have recently enjoyed some advances in terms of non-asymptotic guarantees [9, 71, 6, 73, 77], independent RL has seen less development, with a few exceptions we discuss below.

A recent line of work [65, 56, 45] shows that for zero-sum extensive form games (EFG), independent policy gradient methods can be formulated in the language of *counterfactual regret minimization* [78], and uses this observation to derive convergence guarantees. Unfortunately, for the general zero-sum stochastic games we consider, reducing to an EFG results in exponential blowup in size with respect to horizon.

Arslan and Yüksel [3] introduce an algorithm for learning stochastic games which can be viewed as a 2-timescale method and show convergence (though without rates) in a somewhat different setting from ours. Perolat et al. [58] provide asymptotic guarantees for an independent two-timescale actor-critic method in zero-sum stochastic games with a "simultaneous-move multistage" structure in which each state can only be visited once. Our result is somewhat more general since it works for arbitrary infinite-horizon stochastic games, and is non-asymptotic.

Zhang et al. [76], Bu et al. [10] recently gave global convergence results for policy gradient methods in two-player zero-sum linear-quadratic games. These results show that if the min-player follows policy gradient updates and the max-player follows the best response at each timestep, the min-player will converge to a Nash equilibrium. These results do not satisfy the independence property defined in Section 3, since they follow an inner-loop/outer-loop structure and assume exact access to gradients of the value function. Interestingly, Mazumdar et al. [48] show that for general-sum linear-quadratic games, independent policy gradient methods can fail to converge even locally.

Two concurrent works also develop provable independent learning algorithms for stochastic games. Lee et al. [38] show that the optimistic gradient algorithm obtains linear rates in the full-information and finite-horizon (undiscounted) setting, where the transition probability function $P$ is known and we have exact access to gradients. Their rate depends on the constant in a certain restricted secant inequality; this constant can be arbitrarily small even in the setting of matrix games (i.e., a single state, $\zeta = 1$, and fixed $A, B$), which causes the rate to be arbitrarily slow. In a setting very similar to that of this paper, Bai et al. [7] propose a model-free upper confidence bound-based algorithm, Nash V-learning, which satisfies the independent learning requirement and has near-optimal sample complexity, achieving superior dependence to Theorem 1 on the parameters $S, A, B, \zeta$, as well as no dependence on $C_{\mathcal{G}}$. However, their work has the limitation of only learning non-Markovian policies, whereas the policies learned by 2-timescale SGDA are Markovian (i.e., only depend on the current state).

**Minimax optimization and (non-monotone) variational inequalities.**  Since the objective $V_\rho(x, y)$ is continuous, a natural approach to minimizing it is to appeal to black-box algorithms for nonconvex-nonconcave minimization, and more broadly non-montone variational inequalities. In particular, the gradient dominance condition implies that all first-order stationary points are Nash equilibria. Unfortunately, compared to the single-player setting, where many algorithms such as gradient descent find first-order stationary points for arbitrary smooth, nonconvex functions, existing algorithms for non-monotone variational inequalities all require additional assumptions that are not satisfied in our setting. Mertikopoulos et al. [51] give convergence guarantees for non-monotone variational inequalities satisfying the so-called MVI property, which we show fails even for single-state zero-sum stochastic games (Section 5.1). Yang et al. [74] give an alternating gradient descent algorithm which succeeds for nonconvex-nonconcave games under a two-sided Polyak-Łojasiewicz condition, but this condition (which leads to linear convergence) is also not satisfied in our setting. Another complementary line of work develops algorithms for nonconvex-concave problems [59, 68, 46, 55, 36, 43].

# B Proofs from Section 4

## B.1 Additional Notation

**Q-functions and advantage functions.** For policies $\pi_1, \pi_2$, we let $Q^{\pi_1,\pi_2}(s,a,b)$ denote the *Q-value function*:

$$Q^{\pi_1,\pi_2}(s,a,b) := \mathbb{E}_{\pi_1,\pi_2}\left[\sum_{t=0}^T R(s_t,a_t,b_t)|s_0 = s, a_0 = a, b_0 = b\right],$$

and we let $A^{\pi_1,\pi_2}(s,a,b) = Q^{\pi_1,\pi_2}(s,a,b) - V_s(\pi_1,\pi_2)$ denote the *advantage function*.

Throughout this section we abbreviate $V_\rho(x,y) = V_\rho(\pi_x,\pi_y)$, where $\pi_x$ and $\pi_y$ use the $\varepsilon$-greedy direct parameterization in Assumption 1.

## B.2 Full Version of Theorem 1 and Proof

The full version of Theorem 1 is as follows.

**Theorem 1a.** *Let $\epsilon > 0$ be given. Suppose both players follow the independent policy gradient scheme (3) with the parametrization in Assumption 1. If the learning rates satisfy $\eta_x = \Theta\left(\frac{\epsilon_0^{10.5}\zeta^{44.5}}{C_\mathcal{G}^{15.5}(A\vee B)^{9.75}S^{0.75}}\right)$ and $\eta_y = \Theta\left(\frac{\epsilon^6\zeta^{27}}{C_\mathcal{G}^9(A\vee B)^6\sqrt{S}}\right)$ and $\varepsilon_x = \Theta\left(\frac{\zeta^3\cdot\epsilon}{\sqrt{S}\sqrt{A\vee B}C_\mathcal{G}}\right)$ and $\varepsilon_y = \Theta\left(\frac{\zeta^8\epsilon^2}{C_\mathcal{G}^3(A\vee B)\sqrt{S}}\right)$, then we are guaranteed that*

$$\mathbb{E}\left[\frac{1}{N}\sum_{i=1}^N \max_{\pi_2} V_\rho(\pi_{x^{(i)}},\pi_2)\right] - \min_{\pi_1}\max_{\pi_2} V_\rho(\pi_1,\pi_2) \le \epsilon \tag{13}$$

*after $N = O\left(\frac{(A\vee B)^{10.75}S^{1.25}C_\mathcal{G}^{17.5}}{\epsilon^{12.5}\zeta^{48.5}}\right)$ episodes.*

**Proof of Theorem 1a.** This result is essentially an immediate consequence of Theorem 2a. As a first step, we observe that $D_\mathcal{X}, D_\mathcal{Y} \le \sqrt{S}$. Next, we observe that from Lemma 4, both players have $\ell := \frac{4A\vee B}{\zeta^3}$-jointly-Lipschitz gradients:

$$\|\nabla_x V_\rho(x,y) - \nabla_x V_\rho(x',y')\|_2 \le \frac{4(1-\zeta)A}{\zeta^3}\|(x,y) - (x',y')\|_2,$$

$$\|\nabla_y V_\rho(x,y) - \nabla_y V_\rho(x',y')\|_2 \le \frac{4(1-\zeta)B}{\zeta^3}\|(x,y) - (x',y')\|_2.$$

Similarly, by Proposition 3, the function value $V_\rho(x,y)$ is $L := \frac{2\sqrt{A\vee B}}{\zeta^2}$-Lipschitz:

$$\|\nabla_x V_\rho(x,y)\| \le \frac{\sqrt{A}}{\zeta^2}, \quad \text{and} \quad \|\nabla_y V_\rho(x,y)\| \le \frac{\sqrt{B}}{\zeta^2}.$$

Note that $L/\ell \le 1$ since $A \wedge B \wedge 1/\zeta \ge 1$. Lemma 2 guarantees that both players have bounded variance:

$$\mathbb{E}_{\pi_x,\pi_y}\left\|\widehat{\nabla}_x - \nabla_x V_\rho(x,y)\right\|^2 \le 24\frac{A^2}{\varepsilon_x\zeta^4}, \quad \text{and} \quad \mathbb{E}_{\pi_x,\pi_y}\left\|\widehat{\nabla}_y - \nabla_y V_\rho(x,y)\right\|^2 \le 24\frac{B^2}{\varepsilon_y\zeta^4}. \tag{14}$$

Lemma 1a guarantees that the gradient domination conditions 2 and 3 of Assumption 2 are satisfied with $\mu_y = \mu_x = \zeta/C_\mathcal{G}$, and additive components $\varepsilon_x, \varepsilon_y$ equal to $\frac{2\varepsilon_x}{\zeta^2}$ and $\frac{2\varepsilon_y}{\zeta^2}$, respectively:

$$\frac{\zeta}{C_\mathcal{G}}\left(V_\rho(x,y) - \min_{x'} V_\rho(x',y)\right) - \frac{2\varepsilon_x}{\zeta^2} \le \max_{\bar{x}\in\Delta(\mathcal{A})^{|\mathcal{S}|}}\langle\nabla_x V_\rho(x,y), x - \bar{x}\rangle,$$

and

$$\frac{\zeta}{C_\mathcal{G}}\left(\max_{y'} V_\rho(x,y') - V_\rho(x,y)\right) - \frac{2\varepsilon_y}{\zeta^2} \le \max_{\bar{y}\in\Delta(\mathcal{B})^{|\mathcal{S}|}}\langle\nabla_y V_\rho(x,y), \bar{y} - y\rangle.$$

Note that $L/\ell < 1$. By Theorem 2a, for a desired accuracy level $\epsilon > 0$, if we set:

$$\eta_y = \Theta\left(\frac{\epsilon^4 \zeta^{15}\varepsilon_y}{C_\mathcal{G}^2(A\vee B)^5}\right) = \Theta\left(\frac{\epsilon^4(\zeta/C_\mathcal{G})^2}{((A\vee B)/\zeta^3)^3((A\vee B)/\zeta^4 + B^2/(\varepsilon_y\zeta^4))}\right), \tag{15}$$

$$\eta_x = \Theta\left(\frac{\epsilon^8\zeta^{27}\varepsilon_y\sqrt{\varepsilon_x}}{C_\mathcal{G}^4(A\vee B)^{8.5}}\right) \tag{16}$$

$$= \Theta\left(\frac{\epsilon^8(\zeta/C_\mathcal{G})^4}{((A\vee B)/\zeta^3)^5 \frac{\sqrt{A\vee B}}{\zeta^2}\cdot((A\vee B)/\zeta^4 + B^2/(\varepsilon_y\zeta^4))\sqrt{(A\vee B)/\zeta^4 + A^2/(\varepsilon_x\zeta^4)}}\right),$$

$$N \geq \Omega\left(\frac{\sqrt{S}\cdot\frac{A\vee B}{\zeta^2}}{\epsilon^2\eta_x}\right), \tag{17}$$

we have that

$$\frac{1}{N+1}\sum_{i=0}^{N}\max_y V_\rho(x^{(i)},y) - \min_x\max_y V_\rho(x,y)$$

$$\leq O\left(\frac{\epsilon C_\mathcal{G}}{\zeta} + \frac{\varepsilon_x C_\mathcal{G}}{\zeta^3} + \frac{\sqrt{(A\vee B)/\zeta^3}\sqrt{\varepsilon_y/\zeta^2}(C_\mathcal{G}/\zeta)}{\sqrt{\zeta/C_\mathcal{G}}}\right)$$

$$\leq O\left(\frac{\epsilon C_\mathcal{G}}{\zeta} + \frac{\varepsilon_x C_\mathcal{G}}{\zeta^3} + \frac{\sqrt{\varepsilon_y}C_\mathcal{G}^{1.5}\sqrt{A\vee B}}{\zeta^4}\right). \tag{18}$$

Recall that $x\mapsto\pi_x, y\mapsto\pi_y$ denote the $\varepsilon_x$- and $\varepsilon_y$-greedy parametrizations, respectively (where $\varepsilon_x,\varepsilon_y$ are as in the statement of Theorem 1a). Then for any $\pi_1:\mathcal{S}\to\Delta(\mathcal{A})$ (respectively, $\pi_2:\mathcal{S}\to\Delta(\mathcal{B})$), there is some $x\in\Delta(\mathcal{A})^S$ (respectively, $y\in\Delta(\mathcal{B})^s$) so that $\|\pi_1-\pi_x\|\leq 2\sqrt{S}\varepsilon_x$ (respectively, $\|\pi_2-\pi_y\|\leq 2\sqrt{S}\varepsilon_y$) for each $s\in\mathcal{S},a\in\mathcal{A}$ (respectively, $b\in\mathcal{B}$). Moreover, recall that Proposition 3 shows that the function $(x,y)\mapsto V_\rho(x,y)$ is $L$-Lipschitz for *any* $\varepsilon$-greedy parametrization, in particular for the one given by $\varepsilon_x=\varepsilon_y=0$. Thus,

$$\left|\left(\frac{1}{N+1}\sum_{i=0}^{N}\max_y V_\rho(x^{(i)},y) - \min_x\max_y V_\rho(x,y)\right) - \left(\frac{1}{N+1}\sum_{i=0}^{N}\max_{\pi_2} V_\rho(\pi_{x^{(i)}},\pi_2) - \min_{\pi_1}\max_{\pi_2} V_\rho(\pi_1,\pi_2)\right)\right|$$

$$\leq O\left(\frac{\sqrt{S}(\varepsilon_x\vee\varepsilon_y)\sqrt{A\vee B}}{\zeta^2}\right).$$

From (18) it now follows that

$$\frac{1}{N+1}\sum_{i=0}^{N}\max_{\pi_2} V_\rho(\pi_{x^{(i)}},\pi_2)-\min_{\pi_1}\max_{\pi_2} V_\rho(\pi_1,\pi_2) \leq O\left(\frac{\epsilon C_\mathcal{G}}{\zeta} + \frac{\varepsilon_x C_\mathcal{G}}{\zeta^3} + \frac{\sqrt{\varepsilon_y}C_\mathcal{G}^{1.5}\sqrt{A\vee B}}{\zeta^4} + \frac{\sqrt{S}(\varepsilon_x\vee\varepsilon_y)\sqrt{A\vee B}}{\zeta^2}\right).$$

To achieve a desired accuracy level $\epsilon_0$, it follows from (15), (16), and (17) that if we set

$$\varepsilon_x = \Theta\left(\frac{\zeta^3\cdot\epsilon_0}{\sqrt{S}\sqrt{A\vee B}C_\mathcal{G}}\right), \qquad \varepsilon_y = \Theta\left(\frac{\zeta^8\epsilon_0^2}{C_\mathcal{G}^3(A\vee B)\sqrt{S}}\right), \qquad \epsilon \leq O\left(\frac{\zeta\epsilon_0}{C_\mathcal{G}}\right),$$

and

$$\eta_y = \Theta\left(\frac{\epsilon_0^6\zeta^{27}}{C_\mathcal{G}^9(A\vee B)^6\sqrt{S}}\right) = \Theta\left(\frac{(\zeta\epsilon_0/C_\mathcal{G})^4\zeta^{15}\cdot\frac{\zeta^8\epsilon_0^2}{C_\mathcal{G}^3(A\vee B)\sqrt{S}}}{C_\mathcal{G}^2(A\vee B)^5}\right),$$

$$\eta_x = \Theta\left(\frac{\epsilon_0^{10.5}\zeta^{44.5}}{C_\mathcal{G}^{15.5}(A\vee B)^{9.75}S^{0.75}}\right) = \Theta\left(\frac{(\zeta\epsilon_0/C_\mathcal{G})^8\zeta^{27}\left(\frac{\zeta^8\epsilon_0^2}{C_\mathcal{G}^3(A\vee B)\sqrt{S}}\right)\sqrt{\frac{\zeta^3\cdot\epsilon_0}{\sqrt{S}\sqrt{A\vee B}C_\mathcal{G}}}}{C_\mathcal{G}^4(A\vee B)^{8.5}}\right),$$

$$N = \Omega\left(\frac{(A\vee B)^{10.75}S^{1.25}C_\mathcal{G}^{17.5}}{\epsilon_0^{12.5}\zeta^{48.5}}\right) \geq \Omega\left(\frac{\sqrt{S}\cdot\frac{A\vee B}{\zeta^2}}{(\zeta\epsilon_0/C_\mathcal{G})^2\eta_x}\right),$$

then we have

$$\frac{1}{N+1}\sum_{i=0}^{N}\max_{\pi_2}V_\rho(\pi_{x^{(i)}},\pi_2)-\min_{\pi_1}\max_{\pi_2}V_\rho(\pi_1,\pi_2)\leq\epsilon_0.$$

$\square$

## B.3 Proofs for Additional Results

**Proof of Proposition 1.** We define the following game $\mathcal{G}$ with state space $\mathcal{S}:=\{1,2,3,4,5\}$, action spaces $\mathcal{A}=\mathcal{B}=\{0,1\}$, and any stopping probability $\zeta>0$.

The transitions and rewards are as follows:

- In state 1, with probability $\zeta$, the game stops. Conditioned on not stopping:
  - If actions $(0,0)$ are taken, the game moves to state 2.
  - If actions $(0,1)$ are taken, the game moves to state 3.
  - If actions $(1,0)$ are taken, the game moves to state 4.
  - If actions $(1,1)$ are taken, the game moves to state 5.

  Both players receive 0 reward in state 1.

- In state $2\leq i\leq 5$, the game stops with probability $\zeta$, and otherwise always moves back to state 1. Furthermore, player 1 receives reward $i-1$ and player 2 receives reward $1-i$ (regardles of their actions).

Let the initial state distribution $\rho=\delta_1$ be defined by

$$\rho(1)=\rho(2)=\rho(4)=\rho(5)=1/4,\quad\text{and}\quad\rho(3)=0.$$

Clearly, the value $V_\rho(\pi_1,\pi_2)$ of the game depends only on the policies at state 1, i.e., $\pi_1(\cdot|1),\pi_2(\cdot|1)$. If $\pi_1(0|1)=\pi_2(1|1)=1$, then certainly $d_\rho^{\pi_1,\pi_2}(3)>0$, and therefore $\max_{\pi_1,\pi_2}\left\|\frac{d_\rho^{\pi_1,\pi_2}}{\rho}\right\|_\infty$ is infinite.

On the other hand, let us now consider the best-response policies:

- For any policy $\pi_1$ of the min-player, all policies $\pi_2\in\Pi_2^\star(\pi_1)$ of the max-player satisfy $\pi_2(0|1)=1$. This follows since player 2 prefers state 2 to state 3, and state 4 to state 5.

  In particular, for any pair $(\pi_1,\pi_2)$ with $\pi_2\in\Pi_2^\star(\pi_1)$, we have that $d_\rho^{\pi_1,\pi_2}(3)=0$.

- For any policy $\pi_2$ of the max-player, all policies $\pi_1\in\Pi_1^\star(\pi_1)$ of the min-player satisfy $\pi_1(1|0)=1$. This follows since player 1 prefers state 4 to state 2, and state 5 to state 3.

  In particular, for any pair $(\pi_1,\pi_2)$ with $\pi_1\in\Pi_1^\star(\pi_2)$, we have that $d_\rho^{\pi_1,\pi_2}(3)=0$.

It follows that

$$C_\mathcal{G}=\max\left\{\max_{\pi_2}\min_{\pi_1\in\Pi_1^\star(\pi_2)}\left\|\frac{d_\rho^{\pi_1,\pi_2}}{\rho}\right\|_\infty,\max_{\pi_1}\min_{\pi_2\in\Pi_2^\star(\pi_1)}\left\|\frac{d_\rho^{\pi_1,\pi_2}}{\rho}\right\|_\infty\right\}<\infty,$$

which completes the proof of the proposition. $\square$

## B.4 Supporting Lemmas

**Lemma 2.** Suppose that players follow that $\varepsilon$-greedy direct parameterization in Assumption 1 with parameters $\varepsilon_x$ and $\varepsilon_y$. Given parameters $x\in\Delta(\mathcal{A})^{|\mathcal{S}|},y\in\Delta(\mathcal{A})^{|\mathcal{S}|}$ suppose the players estimate their gradients using the REINFORCE estimator:

$$\widehat{\nabla}_x:=R_T\sum_{t=0}^{T}\nabla\log\pi_x(a_t\mid s_t),\quad\widehat{\nabla}_y:=R_T\sum_{t=0}^{T}\nabla\log\pi_y(b_t\mid s_t),\tag{19}$$

under trajectories obtained by following $\pi_x$ and $\pi_y$. Then we have

$$\mathbb{E}_{\pi_x,\pi_y}\widehat{\nabla}_x=\nabla_xV_\rho(\pi_x,\pi_y),\qquad\mathbb{E}_{\pi_x,\pi_y}\widehat{\nabla}_y=\nabla_yV_\rho(\pi_x,\pi_y),\tag{20}$$

and

$$\mathbb{E}_{\pi_x,\pi_y}\left\|\widehat{\nabla}_x-\nabla_xV_\rho(x,y)\right\|^2\leq24\frac{A^2}{\varepsilon_x\zeta^4},\quad\text{and}\quad\mathbb{E}_{\pi_x,\pi_y}\left\|\widehat{\nabla}_y-\nabla_yV_\rho(x,y)\right\|^2\leq24\frac{B^2}{\varepsilon_y\zeta^4}.\tag{21}$$

**Proof of Lemma 2.** We carry the calculation out for the $x$ player, as the $y$ player follows an identical argument. We start by proving that the gradient estimator is unbiased (i.e., (20)). Let $\mathcal{T}$ denote the (infinite) set of all possible trajectories, and for a trajectory $\tau = (s_t, a_t, b_t, r_t)_{0 \le t \le T}$, in $\mathcal{T}$, let $R(\tau) := \sum_{t=0}^{T} r_t$ denote the total reward associated with $\tau$, and for policies $\pi_1, \pi_2$, let

$$\mathrm{Pr}^{\pi_1, \pi_2}(\tau) := \prod_{t=0}^{T} \pi_1(a_t|s_t)\pi_2(b_t|s_t)P(s_{t+1}|s_t, a_t, b_t)$$

be the probability of realizing $\tau$. (Here, we let $s_{T+1}$ denote the event that the game stops at time $T$) Let $T(\tau)$ denote the last time step of trajectory $\tau$. Then

$$\nabla_x V_\rho(x, y)$$
$$= \nabla_x \sum_{\tau \in \mathcal{T}} R(\tau) \mathrm{Pr}^{\pi_x, \pi_y}(\tau)$$
$$= \sum_{\tau \in \mathcal{T}} R(\tau) \nabla_x \mathrm{Pr}^{\pi_x, \pi_y}(\tau)$$
$$= \sum_{\tau \in \mathcal{T}} R(\tau) \mathrm{Pr}^{\pi_x, \pi_y}(\tau) \nabla_x \log \mathrm{Pr}^{\pi_x, \pi_y}(\tau)$$
$$= \sum_{\tau \in \mathcal{T}} R(\tau) \mathrm{Pr}^{\pi_x, \pi_y}(\tau) \nabla_x \left( \sum_{t=0}^{T(\tau)} \log \pi_x(a_t|s_t) + \log \pi_y(b_t|s_t) \right)$$
$$= \mathbb{E}_{\pi_x, \pi_y} \left[ \left( \sum_{t=0}^{T} r_t \right) \sum_{t=0}^{T} \nabla_x \log \pi_x(a_t|s_t) \right]$$
$$= \mathbb{E}_{\pi_x, \pi_y} \left[ \widehat{\nabla}_x \right].$$

A similar calculation shows that $\mathbb{E}_{\pi_x, \pi_y} \left[ \widehat{\nabla}_y \right] = \nabla_y V_\rho(x, y)$.

We proceed to bound the variance of the gradient estimator (i.e., establish (21)). Since the gradient estimator is unbiased, we have

$$\mathbb{E}_{\pi_x, \pi_y} \left\| \widehat{\nabla}_x - \nabla_x V_\rho(x, y) \right\|^2 \le \mathbb{E}_{\pi_x, \pi_y} \left\| \widehat{\nabla}_x \right\|^2 \le \mathbb{E}_{\pi_x, \pi_y} \left\| R_T \sum_{t=0}^{T} \nabla \log \pi_x(a_t \mid s_t) \right\|^2.$$

Next, we have

$$\mathbb{E}_{\pi_x, \pi_y} \left\| R_T \sum_{t=0}^{T} \nabla \log \pi_x(a_t \mid s_t) \right\|^2 \le \mathbb{E}_{\pi_x, \pi_y} \left[ (T+1)^2 \left\| \sum_{t=0}^{T} \nabla \log \pi_x(a_t \mid s_t) \right\|^2 \right]$$
$$\le \mathbb{E}_{\pi_x, \pi_y} \left[ (T+1)^3 \sum_{t=0}^{T} \| \nabla \log \pi_x(a_t \mid s_t) \|^2 \right]$$
$$= \mathbb{E}_{\pi_x, \pi_y} \left[ (T+1)^3 \sum_{t=0}^{T} \sum_{s,a} (1 - \varepsilon_x)^2 \mathbb{1}\{s = s_t, a = a_t\} \frac{1}{\pi_x^2(a \mid s)} \right],$$
$$\le \mathbb{E}_{\pi_x, \pi_y} \left[ (T+1)^3 \sum_{t=0}^{T} \sum_{s,a} \mathbb{1}\{s = s_t, a = a_t\} \frac{1}{\pi_x^2(a \mid s)} \right],$$

where the equality is a consequence of the direct parameterization. We further simplify as

$$\mathbb{E}_{\pi_x, \pi_y} \left[ (T+1)^3 \sum_{t=0}^{T} \sum_{s,a} \mathbb{1}\{s = s_t, a = a_t\} \frac{1}{\pi_x^2(a \mid s)} \right] = \sum_{t=0}^{\infty} \sum_{s,a} \mathbb{E}_{\pi_x, \pi_y} \left[ \mathbb{1}_{t \le T} \mathbb{1}\{s = s_t, a = a_t\} \frac{1}{\pi_x^2(a \mid s)} (T+1)^3 \right]$$
$$= \sum_{t=0}^{\infty} \sum_{s,a} \mathbb{E}_{\pi_x, \pi_y} \left[ \mathbb{1}_{t \le T} \mathbb{1}\{s = s_t, a = a_t\} \frac{1}{\pi_x^2(a \mid s)} (T+1)^3 \right]$$
$$= \sum_{t=0}^{\infty} \sum_{s,a} \mathbb{E}_{\pi_x, \pi_y} \left[ \mathbb{1}_{t \le T} \mathbb{1}\{s_t = s\} \frac{1}{\pi_x(a \mid s)} (T+1)^3 \right]$$
$$\le \frac{A}{\varepsilon_x} \sum_{t=0}^{\infty} \sum_{s,a} \mathbb{E}_{\pi_x, \pi_y} \left[ \mathbb{1}_{t \le T} \mathbb{1}\{s_t = s\} (T+1)^3 \right]$$
$$\le \frac{A^2}{\varepsilon_x} \mathbb{E}_{\pi_x, \pi_y} \left[ (T+1)^4 \right].$$

To conclude, we observe that

$$\mathbb{E}_{\pi_x,\pi_y}\left[(T+1)^4\right] \le \sum_{t=0}^{\infty}(1-\zeta)^t\zeta(t+1)^4 = \frac{\zeta}{1-\zeta}\sum_{t=1}^{\infty}(1-\zeta)^t t^4 \le \frac{24}{\zeta^4}.$$

$\square$

Define, for any $s_0 \in \mathcal{S}$ and policies $\pi_1, \pi_2$,

$$\tilde{d}_{s_0}^{\pi_1,\pi_2}(s) := \sum_{t\ge0}\Pr^{\pi_1,\pi_2}(s_t = s|s_0),$$

and $\tilde{d}_{\rho}^{\pi_1,\pi_2}(s) := \mathbb{E}_{s_0\sim\rho}[\tilde{d}_{s_0}^{\pi_1,\pi_2}(s)]$ be the *un-normalized* state visitation distribution. Also let $\Pr^{\pi_1,\pi_2}(\tau|s_0)$ (respectively, $\Pr^{\pi_1,\pi_2}(\tau|\rho)$) be the distribution of trajectories $\tau$ given policies $\pi_1, \pi_1$ and initial state $s_0$ (respectively, initial state distribution $\rho$).

**Proposition 3.** In the direct parameterization with $\varepsilon$-greedy exploration (Assumption 1), we have, for all $s \in \mathcal{S}, a \in \mathcal{A}, b \in \mathcal{B}$,

$$\frac{\partial V_\rho(x,y)}{\partial x_{s,a}} = (1-\varepsilon_x)\tilde{d}_{\rho}^{\pi_x,\pi_y}(s)\mathbb{E}_{b\sim\pi_y(\cdot|s)}\left[Q^{\pi_x,\pi_y}(s,a,b)\right]$$

$$\frac{\partial V_\rho(x,y)}{\partial y_{s,b}} = (1-\varepsilon_y)\tilde{d}_{\rho}^{\pi_x,\pi_y}(s)\mathbb{E}_{a\sim\pi_x(\cdot|s)}\left[Q^{\pi_x,\pi_y}(s,a,b)\right].$$

and so it follows that for all $\varepsilon_x, \varepsilon_y \ge 0$,

$$\left|\frac{\partial V_\rho}{\partial x_{s,a}}(x,y)\right| \le \frac{1}{\zeta}d_{\rho}^{\pi_x,\pi_y}(s)\left|\mathbb{E}_{b\sim y(\cdot|s)}[Q^{\pi_x,\pi_y}(s,a,b)]\right|$$

$$\left|\frac{\partial V_\rho}{\partial y_{s,b}}(x,y)\right| \le \frac{1}{\zeta}d_{\rho}^{\pi_x,\pi_y}(s)\left|\mathbb{E}_{a\sim x(\cdot|s)}[Q^{\pi_x,\pi_y}(s,a,b)]\right|.$$

As a consequence, $\|\nabla_x V_\rho(x,y)\| \le \frac{\sqrt{A}}{\zeta^2}$ and $\|\nabla_y V_\rho(x,y)\| \le \frac{\sqrt{B}}{\zeta^2}$.

**Proof of Proposition 3.** Note that for any $s \in \mathcal{S}$, $\tilde{d}_{\rho}^{\pi_1,\pi_2}(s) \le \frac{d_{\rho}^{\pi_1,\pi_2}(s)}{\zeta}$.

Fix any initial state $s_0 \in \mathcal{S}$. Note that

$\nabla_x V_{s_0}(x,y)$

$= \nabla_x \sum_{a_0\in\mathcal{A}} \pi_x(a_0|s_0)\mathbb{E}_{b_0\sim\pi_y(\cdot|s_0)}[Q^{\pi_x,\pi_y}(s_0,a_0,b_0)]$

$= \sum_{a_0}(\nabla_x\pi_x(a_0|s_0))\mathbb{E}_{b_0\sim\pi_y(\cdot|s_0)}[Q^{\pi_x,\pi_y}(s_0,a_0,b_0)] + \sum_{a_0}\pi_x(a_0|s_0)\mathbb{E}_{b_0\sim\pi_y(\cdot|s_0)}[\nabla_x Q^{\pi_x,\pi_y}(s_0,a_0,b_0)]$

$= \sum_{a_0}\pi_x(a_0|s_0)(\nabla_x\log\pi_x(a_0|s_0))\mathbb{E}_{b_0\sim\pi_y(\cdot|s_0)}[Q^{\pi_x,\pi_y}(s_0,a_0,b_0)]$

$\quad + \sum_{a_0}\pi_x(a_0|s_0)\mathbb{E}_{b_0\sim\pi_y(\cdot|s_0)}\left[\sum_{s_1}P(s_1|s_0,a_0,b_0)\nabla_x V_{s_1}(x,y)\right]$

$= \mathbb{E}_{\tau\sim\Pr^{\pi_x,\pi_y}(\cdot|s_0)}\left[(\nabla_x\log\pi_x(a_0|s_0))Q^{\pi_x,\pi_y}(s_0,a_0,b_0)\right]$

$\quad + \mathbb{E}_{\tau\sim\Pr^{\pi_x,\pi_y}(\cdot|s_0)}\left[\mathbb{1}_{T\ge1}\nabla_x V_{s_1}(x,y)\right].$

Note that the above calculation holds also when $s_0$ is replaced with any distribution $\rho \in \Delta(\mathcal{S})$. It follows by induction and the fact that $\Pr[T \ge t] \le (1-\zeta)^t$ for any $t \ge 0$ that for any $\rho \in \Delta(\mathcal{S})$,

$$\nabla_x V_\rho(x,y) = \mathbb{E}_{\tau\sim\Pr^{\pi_x,\pi_y}(\cdot|s_0)}\left[\sum_{t=0}^{T}(\nabla_x\log\pi_x(a_t|s_t))Q^{\pi_x,\pi_y}(s_t,a_t,b_t)\right]$$

$$= \sum_{s\in\mathcal{S}}\mathbb{E}_{a\sim\pi_x(\cdot|s)}\mathbb{E}_{b\sim\pi_y(\cdot|s)}\left[\tilde{d}_{\rho}^{\pi_x,\pi_y}(s)(\nabla_x\log\pi_x(a|s))Q^{\pi_x,\pi_y}(s,a,b)\right].$$

Thus, for any $s \in \mathcal{S}, a \in \mathcal{A}$, we have

$$\frac{\partial V_\rho(x,y)}{\partial x_{s,a}} = (1-\varepsilon_x)\tilde{d}_{\rho}^{\pi_x,\pi_y}(s)\mathbb{E}_{b\sim\pi_y(\cdot|s)}\left[Q^{\pi_x,\pi_y}(s,a,b)\right],$$

and so it follows that
$$\left|\frac{\partial V_\rho(x,y)}{\partial x_{s,a}}\right| \le \frac{d_\rho^{\pi_x,\pi_y}(s)}{\zeta}\left|\mathbb{E}_{b\sim\pi_y(\cdot|s)}\left[Q^{\pi_x,\pi_y}(s,a,b)\right]\right|.$$
The inequality for the derivative with respect to $y$ follows in a symmetric manner. $\qquad\square$

**Lemma 3** (Performance difference lemma). For all policies $\pi_1, \pi_1', \pi_2, \pi_2'$ and distributions $\rho \in \Delta(\mathcal{S})$,
$$V_\rho(\pi_1,\pi_2) - V_\rho(\pi_1',\pi_2) = \sum_{s\in\mathcal{S}} \tilde{d}_\rho^{\pi_1,\pi_2}(s)\mathbb{E}_{a\sim\pi_1(\cdot|s)}\mathbb{E}_{b\sim\pi_2(\cdot|s)}\left[A^{\pi_1',\pi_2}(s,a,b)\right]$$
$$V_\rho(\pi_1,\pi_2) - V_\rho(\pi_1,\pi_2') = \sum_{s\in\mathcal{S}} \tilde{d}_\rho^{\pi_1,\pi_2}(s)\mathbb{E}_{a\sim\pi_1(\cdot|s)}\mathbb{E}_{b\sim\pi_2(\cdot|s)}\left[A^{\pi_1',\pi_2}(s,a,b)\right].$$

**Proof of Lemma 3.** Note that, for any $s \in \mathcal{S}$,
$$V_s(\pi_1,\pi_2) - V_s(\pi_1',\pi_2)$$
$$= \mathbb{E}_{\tau\sim\Pr^{\pi_1,\pi_2}(\cdot|s)}\left[\sum_{t=0}^T R(s_t,a_t,b_t)\right] - V_s(\pi_1',\pi_2)$$
$$= \mathbb{E}_{\tau\sim\Pr^{\pi_1,\pi_2}(\cdot|s)}\left[\sum_{t=0}^T R(s_t,a_t,b_t) + V_{s_t}(\pi_1',\pi_2) - V_{s_t}(\pi_1',\pi_2)\right] - V_s(\pi_1',\pi_2)$$
$$= \mathbb{E}_{\tau\sim\Pr^{\pi_1,\pi_2}(\cdot|s)}\left[\sum_{t=0}^T R(s_t,a_t,b_t) + \mathbb{1}_{t+1\le T}V_{s_{t+1}}(\pi_1',\pi_2) - V_{s_t}(\pi_1',\pi_2)\right]$$
$$= \mathbb{E}_{\tau\sim\Pr^{\pi_1,\pi_2}(\cdot|s)}\left[\sum_{t=0}^T R(s_t,a_t,b_t) + \mathbb{E}\left[\mathbb{1}_{t+1\le T}V_{s_{t+1}}(\pi_1',\pi_2)|s_t,a_t,b_t\right] - V_{s_t}(\pi_1',\pi_2)\right]$$
$$= \mathbb{E}_{\tau\sim\Pr^{\pi_1,\pi_2}(\cdot|s)}\left[\sum_{t=0}^T A^{\pi_1',\pi_2}(s_t,a_t,b_t)\right]$$
$$= \sum_{s'\in\mathcal{S}} \tilde{d}_s^{\pi_1,\pi_2}(s')\mathbb{E}_{a\sim\pi_1(\cdot|s')}\mathbb{E}_{b\sim\pi_2(\cdot|s')}\left[A^{\pi_1',\pi_2}(s',a,b)\right].$$
The proof of the second inequality in the lemma is symmetric. $\qquad\square$

**Lemma 1a.** Suppose that players follow the $\varepsilon$-greedy direct parameterization of Assumption 1 with parameters $\varepsilon_x$ and $\varepsilon_y$. Then for all $x \in \Delta(\mathcal{A})^{|\mathcal{S}|}$, $y \in \Delta(\mathcal{B})^{|\mathcal{S}|}$ we have
$$V_\rho(x,y) - \min_{x'} V_\rho(x',y) \le \min_{\pi_1\in\Pi_1^*(\pi_y)}\left\|\frac{d_\rho^{\pi_1,\pi_y}}{\rho}\right\|_\infty \left(\frac{1}{\zeta}\max_{\bar{x}\in\Delta(\mathcal{A})^{|\mathcal{S}|}}\langle\nabla_x V_\rho(x,y), x-\bar{x}\rangle + \frac{2\varepsilon_x}{\zeta^3}\right), \quad (22)$$
and
$$\max_{y'} V_\rho(x,y') - V_\rho(x,y) \le \min_{\pi_2\in\Pi_2^*(\pi_x)}\left\|\frac{d_\rho^{\pi_x,\pi_2}}{\rho}\right\|_\infty \left(\frac{1}{\zeta}\max_{\bar{y}\in\Delta(\mathcal{B})^{|\mathcal{S}|}}\langle\nabla_y V_\rho(x,y), \bar{y}-y\rangle + \frac{2\varepsilon_y}{\zeta^3}\right), \quad (23)$$

**Proof of Lemma 1.** We prove the inequality for $x$ player. The inequality for the $y$ player follows by symmetry. For a policy $\pi_y$, let $\pi_1^*(\pi_y) \in \Pi_1^*(\pi_y)$ denote a policy minimizing $\left\|\frac{d_\rho^{\pi_1,\pi_y}}{\rho}\right\|_\infty$ (whose existence follows from compactness of the space of policies).

Using the performance difference lemma, we have
$$V_\rho(x,y) - \min_{x'} V_\rho(x',y)$$
$$\le V_\rho(\pi_x,\pi_y) - V_\rho(\pi_1^*(\pi_y),\pi_y)$$
$$= \sum_{s,a} \tilde{d}_\rho^{\pi_1^*(\pi_y),\pi_y}(s)\pi_1^*(\pi_y)(a\mid s)\,\mathbb{E}_{b\sim\pi_y(\cdot|s)}\left[-A^{\pi_x,\pi_y}(s,a,b)\right]$$
$$\le \sum_s \tilde{d}_\rho^{\pi_1^*(\pi_y),\pi_y}(s)\max_a \mathbb{E}_{b\sim\pi_y(\cdot|s)}\left[-A^{\pi_x,\pi_y}(s,a,b)\right]$$
$$\le \left\|\frac{\tilde{d}_\rho^{\pi_1^*(\pi_y),\pi_y}}{\tilde{d}_\rho^{\pi_x,\pi_y}(s)}\right\|_\infty \sum_s \tilde{d}_\rho^{\pi_x,\pi_y}(s)\max_a \mathbb{E}_{b\sim\pi_y(\cdot|s)}\left[-A^{\pi_x,\pi_y}(s,a,b)\right].$$

We observe that $\left\|\frac{\tilde{d}_\rho^{\pi_1^\star(\pi_y),\pi_y}}{\tilde{d}_\rho^{\pi_x,\pi_y}}\right\|_\infty \le \frac{1}{\zeta}\left\|\frac{d_\rho^{\pi_1^\star(\pi_y),\pi_y}}{\rho}\right\|_\infty \le \frac{1}{\zeta}C_\mathcal{G}$. Next, we have

$$\sum_{s,a}\tilde{d}_\rho^{\pi_x,\pi_y}(s)\max_a\mathbb{E}_{b\sim\pi_y(\cdot|s)}[-A^{\pi_x,\pi_y}(s,a,b)]$$

$$= \max_{\bar{x}\in\Delta(\mathcal{A})^{|\mathcal{S}|}}\sum_{s,a}\tilde{d}_\rho^{\pi_x,\pi_y}(s)\bar{x}_{s,a}\,\mathbb{E}_{b\sim\pi_y(\cdot|s)}[-A^{\pi_x,\pi_y}(s,a,b)]$$

$$= \max_{\bar{x}\in\Delta(\mathcal{A})^{|\mathcal{S}|}}\sum_{s,a}\tilde{d}_\rho^{\pi_x,\pi_y}(s)(\pi_x(a\mid s)-\bar{x}_{s,a})\,\mathbb{E}_{b\sim\pi_y(\cdot|s)}[Q^{\pi_x,\pi_y}(s,a,b)]$$

$$= \max_{\bar{x}\in\Delta(\mathcal{A})^{|\mathcal{S}|}}\sum_{s,a}\tilde{d}_\rho^{\pi_x,\pi_y}(s)((1-\varepsilon_x)x_{s,a}+\varepsilon_x A^{-1}-\bar{x}_{s,a})\,\mathbb{E}_{b\sim\pi_y(\cdot|s)}[Q^{\pi_x,\pi_y}(s,a,b)],$$

$$\le \max_{\bar{x}\in\Delta(\mathcal{A})^{|\mathcal{S}|}}\sum_{s,a}\tilde{d}_\rho^{\pi_x,\pi_y}(s)((1-\varepsilon_x)x_{s,a}+\varepsilon_x A^{-1}-\varepsilon_x\bar{x}_{s,a}-(1-\varepsilon_x)\bar{x}_{s,a})\,\mathbb{E}_{b\sim\pi_y(\cdot|s)}[Q^{\pi_x,\pi_y}(s,a,b)]$$

$$\le (1-\varepsilon_x)\max_{\bar{x}\in\Delta(\mathcal{A})^{|\mathcal{S}|}}\sum_{s,a}\tilde{d}_\rho^{\pi_x,\pi_y}(s)(x_{s,a}-\bar{x}_{s,a})\,\mathbb{E}_{b\sim\pi_y(\cdot|s)}[Q^{\pi_x,\pi_y}(s,a,b)]+\frac{2\varepsilon_x}{\zeta^2}$$

$$= \max_{\bar{x}\in\Delta(\mathcal{A})^{|\mathcal{S}|}}\langle\nabla_x V_\rho(x,y),x-\bar{x}\rangle+\frac{2\varepsilon_x}{\zeta^2}, \tag{24}$$

where (24) follows from Proposition 3. Rearranging, this establishes that

$$V_\rho(x,y)-V_\rho(x^\star(y),y)\le\left\|\frac{d_\rho^{\pi_1^\star(\pi_y),\pi_y}}{\rho}\right\|_\infty\left(\frac{1}{\zeta}\max_{\bar{x}\in\Delta(\mathcal{A})^{|\mathcal{S}|}}\langle\nabla_x V_\rho(x,y),x-\bar{x}\rangle+\frac{2\varepsilon_x}{\zeta^3}\right).$$

$\square$

The following lemma, which is a consequence of Lemma E.3 of Agarwal et al. [2], establishes that the direct parameterization leads to Lipschitz gradients.

**Lemma 4** (Smoothness). For all starting states $s_0$, and for all policies $x,x',y,y'$, it holds that

$$\|\nabla_x V_{s_0}(x,y)-\nabla_x V_{s_0}(x',y')\|_2\le\frac{4(1-\zeta)A}{\zeta^3}\|(x,y)-(x',y')\|_2,$$

$$\|\nabla_y V_{s_0}(x,y)-\nabla_y V_{s_0}(x',y')\|_2\le\frac{4(1-\zeta)A}{\zeta^3}\|(x,y)-(x',y')\|_2.$$

## C  Two-Timescale SGDA

### C.1  Algorithm and Main Theorem

Throughout this section we will consider compact and convex subsets $\mathcal{X}\subset\mathbb{R}^{d_x},\mathcal{Y}\subset\mathbb{R}^{d_y}$ of Euclidean space. Our goal will be to find approximate equilibria for the game

$$\min_{x\in\mathcal{X}}\max_{y\in\mathcal{Y}}f(x,y),$$

where $f:\mathcal{X}\times\mathcal{Y}\to\mathbb{R}$ is a continuously differentiable function. We assume that we can only access $f$ through a *stochastic first-order oracle* (Assumption 3), and we analyze a two-timescale version of simultaneous gradient descent-ascent (SGDA) in this model. Before stating the algorithm, we state our regularity assumptions on the function $f$ and the oracle.

Define the *max* function $\Phi:\mathcal{X}\to\mathbb{R}$ and the *min* function $\Psi:\mathcal{Y}\to\mathbb{R}$ as follows:

$$\Phi(x):=\max_{y\in\mathcal{X}}f(x,y),\qquad\Psi(y):=\min_{x\in\mathcal{X}}f(x,y).$$

Moreover, let $D_\mathcal{X}$ denote the diameter of $\mathcal{X}$ and $D_\mathcal{Y}$ denote the diameter of $\mathcal{Y}$. We make the following assumptions about $f(x,y)$. To state the assumption, let $y^\star(x)\in\arg\max_{y\in\mathcal{Y}}f(x,y)$ and $x^\star(y)\in\arg\min_{x\in\mathcal{X}}f(x,y)$ denote arbitrary best-response functions for the $y$ and $x$ players, respectively.

**Assumption 2.** Assume that $\mathcal{X} \subset \mathbb{R}^{d_x}, \mathcal{Y} \subset \mathbb{R}^{d_y}$ are closed and compact subsets of Euclidean space and $f : \mathcal{X} \times \mathcal{Y} \to \mathbb{R}$. We assume that $f$ satisfies:

1. $f$ is $\ell$-smooth and $L$-Lipschitz.

2. For some constants $\varepsilon_y \geq 0, \mu_y > 0$, for each $x \in \mathcal{X}$, the function $y \mapsto f(x, y)$ satisfies the following gradient domination condition:
$$\max_{\bar{y} \in \mathcal{Y}: \|\bar{y} - y\| \leq 1} \langle \bar{y} - y, \nabla_y f(x, y) \rangle \geq \mu_y \cdot (f(x, y^*(x)) - f(x, y)) - \varepsilon_y.$$

3. For some constants $\varepsilon_x \geq 0, \mu_x > 0$, for each $y \in \mathcal{X}$, the function $x \mapsto f(x, y)$ satisfies the following gradient domination condition:
$$\max_{\bar{x} \in \mathcal{X}, \|\bar{x} - x\| \leq 1} \langle x - \bar{x}, \nabla_x f(x, y) \rangle \geq \mu_x \cdot (f(x, y) - f(x^*(y), y)) - \varepsilon_x.$$

**Remark 1** (Empty interior). *If the interior of $\mathcal{X} \times \mathcal{Y} \subset \mathbb{R}^{d_x + d_y}$, denoted by $(\mathcal{X} \times \mathcal{Y})^\circ$, is empty (which is the case for the direct parametrization of policies in Markov games), then in order to ensure that $\nabla f(x, y)$ is well-defined on $\mathcal{X} \times \mathcal{Y}$, we make the technical assumption that $f$ is continuously differentiable on a closed neighborhood $\tilde{\mathcal{X}} \times \tilde{\mathcal{Y}}$, where $\mathcal{X} \subset\subset \tilde{\mathcal{X}}, \mathcal{Y} \subset\subset \tilde{\mathcal{Y}}$[6], which may be assumed without loss of generality to be convex. It is straightforward to check that this assumption holds in our application to Markov games.*

*Suppose $f$ satisfies Assumption 2. In the event that the interior $(\mathcal{X} \times \mathcal{Y})^\circ$ is empty, by compactness of $\mathcal{X}$ and $\mathcal{Y}$, for any $\delta > 0$, there are closed convex neighborhoods $\mathcal{X}_\delta, \mathcal{Y}_\delta$ with $\mathcal{X} \subset\subset \mathcal{X}_\delta, \mathcal{Y} \subset\subset \mathcal{Y}_\delta$, so that any point in $\mathcal{X}_\delta \times \mathcal{Y}_\delta$ is at most distance $\delta$ from a point in $\mathcal{X} \times \mathcal{Y}$, $f$ is $(\ell + \delta)$-smooth and $(L + \delta)$-Lipschitz on $\mathcal{X}_\delta \times \mathcal{Y}_\delta$, and items 2 and 3 hold for any $x \in \mathcal{X}_\delta, y \in \mathcal{Y}_\delta$ with constants $\varepsilon_y + \delta$, $\mu_y - \delta, \mu_x - \delta,$ and $\varepsilon_x + \delta$. We will use this fact in the proof of Lemma 12.*

We formalize the stochastic first-order oracle model our algorithm works in as follows. In this section only, we will denote the iterates of stochastic gradient descent-ascent using $x_t, y_t$ (as opposed to previous sections where we wrote $x^{(i)}, y^{(i)}$).

Given a random variable $\xi \in \Xi$ with law $\mathbb{P}$ (for some sample space $\Xi$), a *stochastic first-order oracle* $G : \mathcal{X} \times \mathcal{Y} \times \Xi \to \mathbb{R}^{d_x + d_y}$ satisfies the following properties.

**Assumption 3** (Stochastic first-order oracle). For variance parameters $\sigma_x, \sigma_y > 0$, the stochastic oracle $G(x, y, \xi) = (G_x(x, y, \xi), G_y(x, y, \xi))$ satisfies:
$$\mathbb{E}[G(x, y, \xi)] = \nabla f(x, y),$$
$$\mathbb{E}[\|G_x(x, y, \xi) - \nabla_x f(x, y, \xi)\|^2] \leq \sigma_x^2,$$
$$\mathbb{E}[\|G_y(x, y, \xi) - \nabla_y f(x, y, \xi)\|^2] \leq \sigma_y^2.$$

Given the stochastic first-order oracle $G = (G_x, G_y)$, the *two-timescale stochastic simultaneous GDA algorithm* (or *SGDA*) draws a sample $\xi_{t-1} \sim \mathbb{P}$, and performs the updates
$$x_t \leftarrow \mathcal{P}_{\mathcal{X}} (x_{t-1} - \eta_x G_x(x_{t-1}, y_{t-1}, \xi_{t-1})), \tag{25}$$
$$y_t \leftarrow \mathcal{P}_{\mathcal{Y}} (y_{t-1} + \eta_y G_y(x_{t-1}, y_{t-1}, \xi_{t-1})). \tag{26}$$

**Main theorem.** Our main theorem for SGDA, Theorem 2a (the full version of Theorem 2), shows that if the learning rate $\eta_x$ of two-timescale SGDA is chosen sufficiently small relative to $\eta_y$, the iterates $x_t$ will approach, on average, the optimal point $x^*$.

For simplicity of presentation, we make the following assumptions regarding the various parameters: $\min\{L, \ell, \sigma_x, \sigma_y, 1/\mu_x, 1/\mu_y\} \geq 1$. These assumptions are essentially without loss of generality (at the cost of potentially worse bounds), since $L, \ell, \sigma_x, \sigma_y, 1/\mu_x, 1/\mu_y$ are *upper bounds* on various properties of the function $f$ and the gradient oracle $G$. Finally, let $\Phi_{1/2\ell}$ denote the *Moreau envelope* of $\Phi$ with parameter $1/2\ell$ (see Appendix C.2).

**Theorem 2a.** *Suppose that Assumption 2 and Assumption 3 hold. For any $\epsilon \in (0,1)$, for two-timescale SGDA with $\eta_y = \Theta\left(\frac{\epsilon^4 \mu_y^2}{\ell^3(L^2+\sigma_y^2)(L/\ell+1)^2}\right), \eta_x = \Theta\left(\frac{\epsilon^8 \mu_y^4}{\ell^5 L(L/\ell+1)^4(L^2+\sigma_y^2)\sqrt{L^2+\sigma_x^2}} \wedge \frac{\epsilon^2}{\ell(L^2+\sigma_x^2)}\right)$, we have*

$$\frac{1}{T+1}\sum_{t=0}^{T} \mathbb{E}\left\|\nabla\Phi_{1/2\ell}(x_t)\right\| \le \epsilon + \sqrt{\frac{8\ell\varepsilon_y}{\mu_y}},$$

*and*

$$\frac{1}{T+1}\sum_{t=0}^{T} \mathbb{E}[\Phi(x_t)] - \Phi(x^\star) \le \left(\frac{1}{\mu_x} + \frac{L}{2\ell}\right) \cdot \left(\epsilon + \sqrt{\frac{8\ell\varepsilon_y}{\mu_y}}\right) + \frac{\varepsilon_x}{\mu_x},$$

*for $T \ge \Omega\left(\frac{(D_{\mathcal{X}}+D_{\mathcal{Y}})L}{\epsilon^2 \eta_x}\right)$.*

To interpret the parameter settings in Theorem 2a, note that if $\varepsilon_x = \varepsilon_y = 0$ and $\sigma_x, \sigma_y, L, \ell, \mu_x, \mu_y, D_{\mathcal{X}}$, and $D_{\mathcal{Y}}$ are all viewed as constants, then if we set $\eta_y \asymp \epsilon^4, \eta_x \asymp \epsilon^8$, we are guaranteed to find an $\epsilon$-suboptimal point within $T \asymp \epsilon^{-10}$ iterations.

## C.2 Technical Preliminaries for Proof

**Non-smooth minimization in the constrained setting.** A function $\varphi : \mathcal{X} \to \mathbb{R}$ is defined to be $\ell$-*weakly convex* if $x \mapsto \varphi(x) + \frac{\ell}{2}\|x\|^2$ is convex. In such a case, we may extend $\varphi$ to a function $\varphi : \mathbb{R}^{d_x} \to \mathbb{R} \cup \{\infty\}$, by $\varphi(x) = \infty$ for $x \notin \mathcal{X}$, and the extended function $\varphi$ remains $\ell$-weakly convex. For a $\ell$-weakly convex function $\varphi$ and $x \in \mathbb{R}^{d_x}$, the *subgradient* of $\varphi$ at $x$ may be defined in terms of the subgradient of the convex function $\tilde{\varphi}(x) := \varphi(x) + \frac{\ell}{2}\|x\|^2$:

$$\partial\varphi(x) := \partial\tilde{\varphi}(x) - \ell x. \tag{27}$$

For any $\lambda > 0$, the *Moreau envelope* $\varphi_\lambda : \mathbb{R}^{d_x} \to \mathbb{R}$ and *proximal map* $\mathrm{prox}_{\lambda\varphi} : \mathbb{R}^{d_x} \to \mathcal{X}$ of $\varphi$ are defined, respectively as follows [21]:

$$\varphi_\lambda(x) := \min_{x'\in\mathcal{X}}\left\{\varphi(x') + \frac{1}{2\lambda}\|x' - x\|^2\right\} \tag{28}$$

$$\mathrm{prox}_{\lambda\varphi}(x) := \arg\min_{x'\in\mathcal{X}}\left\{\varphi(x') + \frac{1}{2\lambda}\|x' - x\|^2\right\}. \tag{29}$$

Let $\Phi(x) = \max_{y\in\mathcal{Y}} f(x,y)$ and $x^* \in \arg\min_{x\in\mathcal{X}} \Phi(x)$.

**Lemma 5** (Lin et al. [43]). *Suppose $f : \mathcal{X} \times \mathcal{Y} \to \mathbb{R}$ is $L$-Lipschitz and $\ell$-smooth. Then:*

1. $\Phi(x)$ *is $L$-Lipschitz.*

2. $\Phi(x)$ *is $\ell$-weakly convex.*

**Lemma 6** (Davis and Drusvyatskiy [21]). *Suppose $\varphi : \mathcal{X} \to \mathbb{R}$ is $\ell$-weakly convex. Then:*

1. $\nabla\varphi_{1/2\ell}(x) = 2\ell(x - \mathrm{prox}_{\varphi/(2\ell)}(x))$.

2. *If $\|\nabla\varphi_{1/2\ell}(x)\|_2 \le \epsilon$, then there is $\hat{x} \in \mathcal{X}$ so that $\|x - \hat{x}\| \le \epsilon/(2\ell)$ and $\min_{\xi\in\partial\varphi(\hat{x})}\|\xi\| \le \epsilon$.*

3. $\nabla\varphi_{1/2\ell}(\cdot)$ *is $\ell$-Lipschitz.*

The following theorem establishes some fundamental properties of $\Phi(x)$.

**Theorem 3** (Danskin's theorem). *Suppose $\mathcal{X} \subset \mathbb{R}^{d_x}$ is an open subset, $\mathcal{Y} \subset \mathbb{R}^{d_y}$ is compact, and $f : \mathcal{X} \times \mathcal{Y} \to \mathbb{R}$ is continuously differentiable and $\ell$-weakly convex. Then $\Phi(x) := \max_{y\in\mathcal{Y}} f(x,y)$ is $\ell$-weakly convex and*

$$\partial\Phi(x) = \mathrm{conv}\left\{\nabla_x f(x,y) : y \in Y(x)\right\},$$

*where*

$$Y(x) := \left\{y : f(x,y) = \max_{y'\in\mathcal{Y}} f(x,y)\right\}.$$

**Descent lemmas for two-timescale SGDA.** Let $(x_t, y_t)$ denote the iterates of two-timescale SGDA, as in (25) and (26). Define $\Delta_t := \Phi(x_t) - f(x_t, y_t)$.

The following lemma, whose proof relies on item 1 of Lemma 6 was shown in [43]; technically, the proof there was given for the unconstrained case (namely, $\mathcal{X} = \mathbb{R}^{d_x}$) and the case where $y \mapsto f(x, y)$ is concave for each $x$, but the proof holds with minimal modifications to our case. For completeness we give the full proof.

**Lemma 7** (Lin et al. [43], Lemma D.3). For two-timescale SGDA, we have:

$$\mathbb{E}[\Phi_{1/(2\ell)}(x_t)] \le \mathbb{E}[\Phi_{1/(2\ell)}(x_{t-1})] + 2\eta_x \ell \mathbb{E}[\Delta_{t-1}] - \frac{\eta_x}{4} \mathbb{E}[\|\nabla \Phi_{1/(2\ell)}(x_{t-1})\|^2] + \eta_x^2 \ell (L^2 + \sigma_x^2).$$

**Proof of Lemma 7.** Set $\hat{x}_{t-1} := \operatorname{prox}_{\Phi/2\ell}(x_{t-1})$, so that

$$\Phi_{1/2\ell}(x_t) \le \Phi(\hat{x}_{t-1}) + \ell \|\hat{x}_{t-1} - x_t\|^2 \le \Phi_{1/2\ell}(x_{t-1}) + \ell \|\hat{x}_{t-1} - x_t\|^2 - \ell \|\hat{x}_{t-1} - x_{t-1}\|^2. \tag{30}$$

Since $\hat{x}_{t-1} \in \mathcal{X}$ and $x_t = \mathcal{P}_{\mathcal{X}}(x_{t-1} - \eta_x G_x(x_{t-1}, y_{t-1}, \xi_{t-1}))$, we have

$$\|\hat{x}_{t-1} - x_t\|^2 \le \|\hat{x}_{t-1} - (x_{t-1} - \eta_x G_x(x_{t-1}, y_{t-1}, \xi_{t-1}))\|^2$$
$$\le \|\hat{x}_{t-1} - x_{t-1}\|^2 + \|\eta_x G_x(x_{t-1}, y_{t-1}, \xi_{t-1})\|^2 + 2\langle \hat{x}_{t-1} - x_{t-1}, \eta_x G_x(x_{t-1}, y_{t-1}, \xi_{t-1})\rangle.$$

Taking the expectation of both sides gives

$$\mathbb{E}[\|\hat{x}_{t-1} - x_t\|^2]$$
$$\le \mathbb{E}[\|\hat{x}_{t-1} - x_{t-1}\|^2] + \eta_x^2 \mathbb{E}[\|G_x(x_{t-1}, y_{t-1}, \xi_{t-1})\|^2] + 2\langle \hat{x}_{t-1} - x_{t-1}, \eta_x \nabla_x f(x_{t-1}, y_{t-1})\rangle$$
$$\le \mathbb{E}[\|\hat{x}_{t-1} - x_{t-1}\|^2] + \eta_x^2 (L^2 + \sigma^2) + 2\mathbb{E}[\langle \hat{x}_{t-1} - x_{t-1}, \eta_x \nabla_x f(x_{t-1}, y_{t-1})\rangle]. \tag{31}$$

Next, we observe that

$$\langle \hat{x}_{t-1} - x_{t-1}, \nabla_x f(x_{t-1}, y_{t-1})\rangle$$
$$\le f(\hat{x}_{t-1}, y_{t-1}) - f(x_{t-1}, y_{t-1}) + \frac{\ell}{2} \|\hat{x}_{t-1} - x_{t-1}\|^2$$
$$\le \Phi(\hat{x}_{t-1}) - f(x_{t-1}, y_{t-1}) + \frac{\ell}{2} \|\hat{x}_{t-1} - x_{t-1}\|^2$$
$$= \Phi(\hat{x}_{t-1}) + \Delta_{t-1} - \Phi(x_{t-1}) + \frac{\ell}{2} \|\hat{x}_{t-1} - x_{t-1}\|^2$$
$$\le \Delta_{t-1} - \frac{\ell}{2} \|\hat{x}_{t-1} - x_{t-1}\|^2 \le \Delta_{t-1}, \tag{32}$$

where the first inequality above follows since $f$ is $\ell$-smooth, the second inequality follows since $\Phi(\hat{x}_{t-1}) \ge f(\hat{x}_{t-1}, y_{t-1})$, and the final inequality (32)) follows since $\Phi(\hat{x}_{t-1}) + \ell \|\hat{x}_{t-1} - x_{t-1}\|^2 \le \Phi(x_{t-1})$ by definition of $\operatorname{prox}_{\Phi/2\ell}(\cdot)$.

By equations (30), (31), and (32), we get

$$\mathbb{E}[\Phi_{1/2\ell}(x_t)] \le \mathbb{E}[\Phi_{1/2\ell}(x_{t-1})] + \ell \eta_x^2 (L^2 + \sigma^2) + 2\ell \eta_x \langle \hat{x}_{t-1} - x_{t-1}, \nabla_x f(x_{t-1}, y_{t-1})\rangle$$
$$\le \mathbb{E}[\Phi_{1/2\ell}(x_{t-1})] + 2\eta_x \ell \mathbb{E}[\Delta_{t-1}] - \eta_x \ell^2 \mathbb{E}[\|\hat{x}_{t-1} - x_{t-1}\|^2] + \ell \eta_x^2 (L^2 + \sigma^2)$$
$$\le \mathbb{E}[\Phi_{1/2\ell}(x_{t-1})] + 2\eta_x \ell \mathbb{E}[\Delta_{t-1}] - \frac{\eta_x}{4} \mathbb{E}[\|\nabla \Phi_{1/2\ell}(x_{t-1})\|^2] + \ell \eta_x^2 (L^2 + \sigma^2).$$

$\square$

To show that the $y$ player approximately tracks the best response (in terms of value), we make use of a slightly different potential function. To describe the approach, set $\lambda \in (0, 1/\ell)$ to be specified later. Letting $(x_t, y_t)$ be the iterates of two-timescale SGDA, for each $t \ge 0$, let $\phi_{t-1} : \mathcal{Y} \to \mathbb{R}$ be the function $\phi_{t-1}(y) := -f(x_{t-1}, y)$, and set $\psi_{t,\lambda}(y) := -(\phi_{t-1})_\lambda(y)$ to be the negated Moreau envelope of $\phi_{t-1}$ with parameter $\lambda$. Our first lemma states that $\psi_{t,\lambda}$ does not change much from iteration to iteration.

**Lemma 8.** For all $t \ge 1$, and $y \in \mathcal{Y}$, we have

$$|\psi_{t,\lambda}(y) - \psi_{t-1,\lambda}(y)| \le L \cdot \|x_{t-1} - x_t\|.$$

**Proof of Lemma 8.** Note that for any $y \in \mathcal{Y}$,

$$|\psi_{t,\lambda}(y) - \psi_{t-1,\lambda}(y)| = \left| \min_{y' \in \mathcal{Y}} \left\{ \frac{1}{2\lambda} \|y - y'\|^2 - f(x_t, y') \right\} - \min_{y' \in \mathcal{Y}} \left\{ \frac{1}{2\lambda} \|y - y'\|^2 - f(x_{t-1}, y') \right\} \right|.$$

Since for all $y' \in \mathcal{Y}$ we have

$$\left\| \frac{1}{2\lambda} \|y - y'\|^2 - f(x_t, y') - \left( \frac{1}{2\lambda} \|y - y'\|^2 - f(x_{t-1}, y') \right) \right\| \le L\|x_{t-1} - x_t\|,$$

the conclusion follows. $\qquad\square$

Now let $\Gamma_t := \|\nabla \psi_{t,\lambda}(y_t)\|$. The following lemma shows that as long as $\Gamma_t$ stays large, $\psi_{t,\lambda}$ decreases each iteration (up to an error term controlled by the learning rate of the $x$ player).

**Lemma 9.** For two-timescale SGDA, for all $t \ge 0$, as long as $\eta_y \le 1/(2\ell)$ and $\lambda \in (0, 1/\ell)$,

$$\mathbb{E}[\psi_{t,\lambda}(y_t)|\mathcal{F}_{t-1}] \ge \psi_{t-1,\lambda}(y_{t-1}) + \eta_y \lambda (1/\lambda - \ell) \cdot \Gamma_{t-1}^2 - L\eta_x \sqrt{L^2 + \sigma_x^2} - \frac{\eta_y^2(L^2 + \sigma_y^2)}{2\lambda}. \quad (33)$$

**Proof of Lemma 9.** Write $g_{t-1} = G_y(x_{t-1}, y_{t-1}, \xi_{t-1})$. Set

$$\hat{y}_{t-1} := \text{prox}_{\lambda \phi_{t-1}}(y_{t-1}) = \arg\min_{y' \in \mathcal{Y}} \left\{ \frac{1}{2\lambda} \|y_{t-1} - y'\|^2 - f(x_{t-1}, y') \right\}.$$

We next need the following lower bound on $\psi_{t-1,\lambda}(y_t)$ in terms of $\psi_{t-1,\lambda}(y_{t-1})$; this calculation was carried out in Davis and Drusvyatskiy [21, Eqs. (2.4) – (2.6)], but we prove the following self-contained lemma after the conclusion of this proof for completeness.

**Lemma 10** (Davis and Drusvyatskiy [21])**.** For $\lambda \in (0, 1/\ell)$, we have

$$\mathbb{E}\left[\psi_{t-1,\lambda}(y_t)|\mathcal{F}_{t-1}\right] \ge \psi_{t-1,\lambda}(y_{t-1}) + \frac{\eta_y}{\lambda}\left( f(x_{t-1}, \hat{y}_{t-1}) - f(x_{t-1}, y_{t-1}) - \frac{1}{2\lambda}\|y_{t-1} - \hat{y}_{t-1}\|^2 \right) - \frac{\eta_y^2(L^2 + \sigma_y^2)}{2\lambda}.$$

By Lemma 10, we have

$$\mathbb{E}\left[\psi_{t-1,\lambda}(y_t)|\mathcal{F}_{t-1}\right] \ge \psi_{t-1,\lambda}(y_{t-1}) + \frac{\eta_y}{\lambda}\left( f(x_{t-1}, \hat{y}_{t-1}) - f(x_{t-1}, y_{t-1}) - \frac{1}{2\lambda}\|y_{t-1} - \hat{y}_{t-1}\|^2 \right) - \frac{\eta_y^2(L^2 + \sigma_y^2)}{2\lambda}$$

$$\ge \psi_{t-1,\lambda}(y_{t-1}) + \frac{\eta_y}{\lambda} \cdot \left( \lambda^2(1/\lambda - \ell) \cdot \|\nabla \psi_{t-1,\lambda}(y_{t-1})\|^2 \right) - \frac{\eta_y^2(L^2 + \sigma_y^2)}{2\lambda}$$

$$= \psi_{t-1,\lambda}(y_{t-1}) + \eta_y \lambda (1/\lambda - \ell) \cdot \Gamma_{t-1}^2 - \frac{\eta_y^2(L^2 + \sigma_y^2)}{2\lambda},$$

where the second inequality above follows by $\ell$-smoothness of $f$. By Lemma 8, we have

$$\mathbb{E}[\psi_{t-1,\lambda}(y_t) - \psi_{t,\lambda}(y_t)|\mathcal{F}_{t-1}] \le L \cdot \mathbb{E}[\|x_{t-1} - x_t\| | \mathcal{F}_{t-1}]$$
$$\le L \cdot \mathbb{E}[\eta_x \cdot \|G_x(x_{t-1}, y_{t-1}, \xi_{t-1})\| | \mathcal{F}_{t-1}]$$
$$\le L\eta_x \cdot \sqrt{L^2 + \sigma_x^2}.$$

Combining the above displays gives that

$$\mathbb{E}[\psi_{t,\lambda}(y_t)|\mathcal{F}_{t-1}] \ge \psi_{t-1,\lambda}(y_{t-1}) + \eta_y \lambda (1/\lambda - \ell) \cdot \Gamma_{t-1}^2 - L\eta_x \sqrt{L^2 + \sigma_x^2} - \frac{\eta_y^2(L^2 + \sigma_y^2)}{2\lambda}.$$

$$\qquad\square$$

**Proof of Lemma 10.** The proof is exactly the argument in Davis and Drusvyatskiy [21, Eqs. (2.4) – (2.6)] and similar to that used in the proof of Lemma 7, but for completeness we repeat this

calculation using our notation. In the setting of Lemma 9, set $\hat{y}_{t-1} \coloneqq \operatorname{prox}_{-\lambda \cdot \phi_{t-1}}(y_{t-1})$, and $g_{t-1} = G_y(x_{t-1}, y_{t-1}, \xi_{t-1})$. Then

$$
\mathbb{E}\left[-(\phi_{t-1})_\lambda(y_t)|\mathcal{F}_{t-1}\right]
$$

$$
\leq \mathbb{E}\left[-\phi_{t-1}(\hat{y}_{t-1}) + \frac{1}{2\lambda}\|y_t - \hat{y}_{t-1}\|^2|\mathcal{F}_{t-1}\right] \tag{34}
$$

$$
\leq -\phi_{t-1}(\hat{y}_{t-1}) + \frac{1}{2\lambda}\mathbb{E}\left[\|y_{t-1} - \eta_y g_{t-1} - \hat{y}_{t-1}\|^2|\mathcal{F}_{t-1}\right] \tag{35}
$$

$$
= -\phi_{t-1}(\hat{y}_{t-1}) + \frac{1}{2\lambda}\|y_{t-1} - \hat{y}_{t-1}\|^2 + \frac{\eta_y^2}{2\lambda}\mathbb{E}\left[\|g_{t-1}\|^2|\mathcal{F}_{t-1}\right] + \frac{\eta_y}{2\lambda}\mathbb{E}\left[\langle \hat{y}_{t-1} - y_{t-1}, g_{t-1}\rangle|\mathcal{F}_{t-1}\right] \tag{36}
$$

$$
\leq -(\phi_{t-1})_\lambda(y_{t-1}) + \frac{\eta_y}{2\lambda}\mathbb{E}\left[\langle \hat{y}_{t-1} - y_{t-1}, g_{t-1}\rangle|\mathcal{F}_{t-1}\right] + \frac{\eta_y^2(L^2 + \sigma_y^2)}{2\lambda} \tag{37}
$$

$$
= -(\phi_{t-1})_\lambda(y_{t-1}) + \frac{\eta_y}{2\lambda}\langle \hat{y}_{t-1} - y_{t-1}, \nabla_y f(x_{t-1}, y_{t-1})\rangle + \frac{\eta_y^2(L^2 + \sigma_y^2)}{2\lambda} \tag{38}
$$

$$
\leq -(\phi_{t-1})_\lambda(y_{t-1}) + \frac{\eta_y}{2\lambda}\left(-\phi_{t-1}(\hat{y}_{t-1}) + \phi_{t-1}(y_{t-1}) + \frac{1}{2\lambda}\|y_{t-1} - \hat{y}_{t-1}\|^2\right) + \frac{\eta_y^2(L^2 + \sigma_y^2)}{2\lambda}, \tag{39}
$$

where (34) is by the definition of the prox-mapping, (35) is by the definition of projection onto a convex set, (37) is by the definition of the Moreau envelope, (38) holds because $g_{t-1}$ is an unbiased estimator of the gradient, and (39) follows since $f$ is $\ell$-smooth and $\lambda \leq 1/\ell$. $\qquad\square$

### C.3 Proof of Theorem 2a

**Proof of Theorem 2a.** By the fact that $f$ satisfies Assumption 2 and Lemma 11 on the function $y \mapsto \phi_t(y) = -f(x_t, y)$, for any $\lambda \in (0, 1/\ell)$ we have

$$
\begin{aligned}
\Delta_t &\coloneqq f(x_t, y^*(x_t)) - f(x_t, y_t) \\
&\leq \frac{L\lambda + 1}{\mu_y} \cdot \underbrace{\|\nabla \psi_{t,\lambda}(y_t)\|}_{=: \Gamma_t} + \frac{\varepsilon_y}{\mu_y}.
\end{aligned} \tag{40}
$$

We next observe that

$$
\mathbb{E}[\Phi_{1/2\ell}(x_{T+1})]
$$

$$
\leq \mathbb{E}[\Phi_{1/2\ell}(x_0)] + 2\eta_x\ell\left(\sum_{t=0}^{T}\mathbb{E}[\Delta_t]\right) - \frac{\eta_x}{4}\left(\sum_{t=0}^{T}\mathbb{E}\left[\|\nabla\Phi_{1/2\ell}(x_t)\|^2\right]\right) + \eta_x^2\ell(L^2 + \sigma_x^2)(T+1)
$$

$$
\leq \mathbb{E}[\Phi_{1/2\ell}(x_0)] + 2\eta_x\ell\left(\sum_{t=0}^{T}\mathbb{E}\left[\frac{L\lambda + 1}{\mu_y}\|\nabla\psi_{t,\lambda}(y_t)\| + \frac{\varepsilon_y}{\mu_y}\right]\right) - \frac{\eta_x}{4}\left(\sum_{t=0}^{T}\mathbb{E}\left[\|\nabla\Phi_{1/2\ell}(x_t)\|^2\right]\right)
$$

$$
+ \eta_x^2\ell(L^2 + \sigma_x^2)(T+1)
$$

$$
\leq \mathbb{E}[\Phi_{1/2\ell}(x_0)] + \frac{2\eta_x\ell(L\lambda + 1)}{\mu_y}\left(\sqrt{\frac{(D_{\mathcal{X}} + D_{\mathcal{Y}})LT}{\eta_y(1-\lambda\ell)}} + T\sqrt{\frac{L\sqrt{L^2 + \sigma_x^2}\eta_x}{\eta_y(1-\lambda\ell)}} + T\sqrt{\frac{(L^2 + \sigma_y^2)\eta_y}{2\lambda(1-\lambda\ell)}}\right)
$$

$$
+ \frac{2\eta_x\ell(T+1)\varepsilon_y}{\mu_y} - \frac{\eta_x}{4}\left(\sum_{t=0}^{T}\mathbb{E}\left[\|\nabla\Phi_{1/2\ell}(x_t)\|^2\right]\right) + \eta_x^2\ell(L^2 + \sigma_x^2)(T+1),
$$

where the first inequality follows from summing the guarantee of Lemma 7 for $t = 1, 2, \ldots, T+1$, the second inequality comes from (40), and the third inequality comes from Lemma 13. Setting $\lambda = 1/2\ell$

and rearranging gives, for $\Delta_\Phi := \mathbb{E}[\Phi_{1/2\ell}(x_0) - \Phi_{1/2\ell}(x_{T+1})]$,

$$\frac{1}{T+1}\sum_{t=0}^{T}\mathbb{E}\left[\|\nabla\Phi_{1/\ell}(x_t)\|^2\right]$$

$$\leq \frac{4\Delta_\Phi}{T\eta_x} + \frac{8\ell(L/\ell+1)}{\mu_y}\sqrt{\frac{2(D_\mathcal{X}+D_\mathcal{Y})L}{T\eta_y}} + \frac{8\ell(L/\ell+1)}{\mu_y}\sqrt{\frac{2L\sqrt{L^2+\sigma_x^2}\eta_x}{\eta_y}} + \frac{8\ell(L/\ell+1)}{\mu_y}\sqrt{\ell(L^2+\sigma_y^2)\eta_y}$$

$$+ 4\eta_x\ell(L^2+\sigma_x^2) + \frac{8\ell\varepsilon_y}{\mu_y}$$

$$\leq \frac{4D_\mathcal{X}L}{T\eta_x} + \frac{8\ell(L/\ell+1)}{\mu_y}\sqrt{\frac{2(D_\mathcal{X}+D_\mathcal{Y})L}{T\eta_y}} + \frac{8\ell(L/\ell+1)}{\mu_y}\sqrt{\frac{2L\sqrt{L^2+\sigma_x^2}\eta_x}{\eta_y}} + \frac{8\ell(L/\ell+1)}{\mu_y}\sqrt{\ell(L^2+\sigma_y^2)\eta_y}$$

$$+ 4\eta_x\ell(L^2+\sigma_x^2) + \frac{8\ell\varepsilon_y}{\mu_y}.$$

Next, for a sufficiently large constant $C > 0$ and for any $\epsilon > 0$, set

$$\eta_y \leq \frac{\epsilon^4\mu_y^2}{C\ell^3(L^2+\sigma_y^2)(L/\ell+1)^2}$$

$$\eta_x \leq \frac{\epsilon^8\mu_y^4}{C\ell^5 L(L/\ell+1)^4(L^2+\sigma_y^2)\sqrt{L^2+\sigma_x^2}} \wedge \frac{\epsilon^2}{C\ell(L^2+\sigma_x^2)}.$$

Then as long as

$$T \geq \frac{C(D_\mathcal{X}+D_\mathcal{Y})L}{\epsilon^2\eta_x}, \tag{41}$$

as long as $C$ is sufficiently large, we get that

$$\frac{1}{T+1}\sum_{t=0}^{T}\mathbb{E}\left[\|\nabla\Phi_{1/2\ell}(x_t)\|\right] \leq \epsilon.$$

Here we have used that if $T$ is set as in (41), then

$$\frac{\ell(L/\ell+1)}{\mu_y}\sqrt{\frac{(D_\mathcal{X}+D_\mathcal{Y})L}{T\eta_y}} \leq \frac{\ell(L/\ell+1)}{\mu_y}\cdot\frac{\epsilon^3\mu_y}{\sqrt{C}\ell^{3/2}\sqrt{L^2+\sigma_y^2}(L/\ell+1)} \leq \epsilon^2.$$

Finally, the guarantee for function value suboptimality follows by applying Lemma 12.

$\square$

### C.4 Supporting Lemmas

Given a convex set $\mathcal{X} \subset \mathbb{R}^n$ and a point $x \in \mathcal{X}$, the *normal cone* of $\mathcal{X}$ at $x$ is the set

$$N_\mathcal{X}(x) := \{x' \in \mathcal{X} : \langle x', y-x\rangle \leq 0 \ \forall y \in \mathcal{X}\},$$

and the *tangent cone* of $\mathcal{X}$ at $x$ is the set

$$T_\mathcal{X}(x) := \text{cl}\{a\cdot(y-x) : y \in \mathcal{X}, a \geq 0\},$$

where cl denotes closure. It is well-known [60] that for any $v \in N_\mathcal{X}(x)$, for all $u \in T_\mathcal{X}(x)$, we have that $\langle v, u\rangle \leq 0$ (in other words, $N_\mathcal{X}(x)$ is contained in the polar of $T_\mathcal{X}(x)$; in fact, $N_\mathcal{X}(x)$ is equal to the polar of $T_\mathcal{X}(x)$).

**Lemma 11.** Suppose that $\phi : \mathcal{X} \to \mathbb{R}$ is $\ell$-smooth, $L$-Lipschitz, and satisfies the gradient domination condition

$$\max_{\bar{x}\in\mathcal{X},\|\bar{x}-x\|\leq 1}\langle x-\bar{x}, \nabla\phi(x)\rangle \geq \mu\cdot(\phi(x)-\phi(x^*)) - \varepsilon,$$

for some $\varepsilon \geq 0, \mu > 0$. Then for any $\lambda \in (0, 1/\ell)$, the Moreau envelope $\phi_\lambda(\cdot)$ satisfies

$$\|\nabla\phi_\lambda(x)\| \geq \frac{\mu}{L\lambda+1}\cdot(\phi(x)-\phi(x^*)) - \frac{\varepsilon}{L\lambda+1}.$$

**Proof of Lemma 11.** Fix $x \in \mathcal{X}$. Let

$$\hat{x} := \operatorname{prox}_{\lambda\phi}(x) = \arg\min_{x' \in \mathcal{X}} \left\{ \phi(x') + \frac{1}{2\lambda} \|x - x'\|^2 \right\}. \tag{42}$$

The first-order optimality conditions to (42) imply that

$$\nabla\phi(\hat{x}) \in \frac{1}{\lambda} \cdot (\hat{x} - x) + N_{\mathcal{X}}(\hat{x}) \subseteq N_{\mathcal{X}}(\hat{x}) + \frac{1}{\lambda} \|\hat{x} - x\| \cdot B_2(1),$$

where $N_{\mathcal{X}}(\hat{x})$ denotes the normal cone of $\mathcal{X}$ at $\hat{x}$. Since for any $\bar{x} \in \mathcal{X}$, $\bar{x} - \hat{x}$ is in the tangent cone at $\hat{x}$, it follows that

$$\mu \cdot (\phi(\hat{x}) - \phi(x^*)) \le \max_{\bar{x} \in \mathcal{X}, \|\bar{x} - \hat{x}\| \le 1} \langle \hat{x} - \bar{x}, \nabla\phi(\hat{x}) \rangle + \varepsilon \le \frac{1}{\lambda} \cdot \|\hat{x} - x\| + \varepsilon.$$

Note that $\frac{1}{\lambda}(x - \hat{x}) = \nabla\phi_\lambda(x)$. Thus, using also that $\phi$ is $L$-Lipschitz, we arrive at

$$\begin{aligned}
\mu \cdot (\phi(x) - \phi(x^*)) &\le \mu \cdot (\phi(\hat{x}) - \phi(x^*)) + L \cdot \|\hat{x} - x\| \\
&\le \left( L + \frac{1}{\lambda} \right) \|\hat{x} - x\| + \varepsilon \\
&= (L\lambda + 1) \cdot \|\nabla\phi_\lambda(x)\| + \varepsilon.
\end{aligned}$$

$\square$

The next lemma (Lemma 12) shows how to convert an $\epsilon$-approximate stationary point with respect to the Moreau envelope into an approximate minimizer for functions $f$ satisfying Assumption 2.

**Lemma 12.** Suppose that $f$ satisfies the conditions of Assumption 2. Then for all $x \in \mathcal{X}$,

$$\Phi(x) - \Phi(x^*) \le \left( \frac{1}{\mu_x} + \frac{L}{2\ell} \right) \cdot \|\nabla\Phi_{1/2\ell}(x)\| + \frac{\varepsilon_x}{\mu_x}. \tag{43}$$

**Proof of Lemma 12.** We first establish the statement of Lemma 12 for points $x \in \mathcal{X}$ for which $\Phi$ is differentiable at $x$. Suppose $x$ is such a point. Since a convex function is differentiable at a point if and only if its subgradient is a singleton at that point [60, Theorem 25.1], it follows from (27) that $\partial\Phi(x)$ is a single vector, which we denote by $\nabla\Phi(x)$.

We first show that $\Phi(x)$ satisfies the following KL-type inequality (see also [74, Lemma A.3], which shows a similar statement):

$$\|\nabla\Phi(x)\| \ge \mu_x \cdot (\Phi(x) - \Phi(x^*)) - \varepsilon_x. \tag{44}$$

To prove (44), fix any $y \in Y(x)$ (so that $f(x, y) = \Phi(x)$), and note that by item 3 of Assumption 2, we have that

$$\max_{\bar{x} \in \mathcal{X}, \|\bar{x} - x\| \le 1} \langle x - \bar{x}, \nabla_x f(x, y) \rangle \ge \mu_x \cdot (f(x, y) - f(x^*(y), y)) - \varepsilon_x = \mu_x \cdot (\Phi(x) - f(x^*(y), y)) - \varepsilon_x.$$

Note note that since $f(x', y) \le \max_{y' \in \mathcal{Y}} f(x', y')$ for each $x'$,

$$f(x^*(y), y) = \min_{x' \in \mathcal{X}} f(x', y) \le \min_{x' \in \mathcal{X}} \left[ \max_{y' \in \mathcal{Y}} f(x', y') \right] = \Phi(x^*).$$

It follows that

$$\max_{\bar{x} \in \mathcal{X}, \|\bar{x} - x\| \le 1} \langle x - \bar{x}, \nabla_x f(x, y) \rangle \ge \mu_x \cdot (\Phi(x) - f(x^*(y), y)) - \varepsilon_x \ge \mu_x \cdot (\Phi(x) - \Phi(x^*)) - \varepsilon_x. \tag{45}$$

By Danskin's theorem (Theorem 3) we have that $\{\nabla\Phi(x)\} = \partial\Phi(x) = \operatorname{conv}\{\nabla_x f(x, y') : y' \in Y(x)\}$, so $\nabla_x f(x, y) = \nabla\Phi(x)$. From (45) and Cauchy-Schwarz it follows that

$$\|\nabla\Phi(x)\| \ge \max_{\bar{x} \in \mathcal{X}, \|\bar{x} - x\| \le 1} \langle x - \bar{x}, \nabla\Phi(x) \rangle = \max_{\bar{x} \in \mathcal{X}, \|\bar{x} - x\| \le 1} \langle x - \bar{x}, \nabla_x f(x, y) \rangle \ge \mu_x \cdot (\Phi(x) - \Phi(x^*)) - \varepsilon_x,$$

establishing (44).

We proceed with the proof of (43). Let $\iota = \|\nabla\Phi_{1/2\ell}(x)\|$. By item 2 of Lemma 6, there is some $\hat{x} \in \mathcal{X}$ so that $\|\hat{x} - x\| \le \iota/(2\ell)$ and $\inf_{v \in \partial\Phi(\hat{x})} \|v\| \le \iota$. By (44), we have

$$\Phi(\hat{x}) - \Phi(x^*) \le \frac{\iota + \varepsilon_x}{\mu_x}.$$

Item 1 of Lemma 5 gives that $\Phi$ is $L$-Lipschitz, and hence

$$\Phi(x) - \Phi(x^*) \le \frac{\iota + \varepsilon_x}{\mu_x} + L \cdot \|\hat{x} - x\| \le \iota \cdot \left(\frac{1}{\mu_x} + \frac{L}{2\ell}\right) + \frac{\varepsilon_x}{\mu_x}.$$

Next we consider any point $x$ for which $\Phi$ is not differentiable at $x$. In the event that the interior $\mathcal{X}^\circ$ is dense in $\mathcal{X}$, we may apply (27) together with [60, Theorem 25.5] to conclude that the set of points at which $\Phi$ is differentiable is dense in $\mathcal{X}^\circ$, and thus in $\mathcal{X}$. Let $x_k \to x$ be a convergent sequence of points approaching a point $x \in \mathcal{X}$ at which $\Phi(\cdot)$ is differentiable. Then the above argument establishes that for each $k$,

$$\Phi(x_k) - \Phi(x^*) \le \left(\frac{1}{\mu_x} + \frac{L}{2\ell}\right) \cdot \|\nabla\Phi_{1/2\ell}(x_k)\| + \frac{\varepsilon_x}{\mu_x}.$$

By continuity of $\Phi$ (Lemma 5, item 1) and of $\nabla\Phi_{1/2\ell}$ (Lemma 5, item 3), it follows that (43) holds at the point $x$.

Finally, we consider the case that $\mathcal{X}^\circ$ is not dense in $\mathcal{X}$ (e.g., $\mathcal{X}^\circ$ may be empty). In this case we consider the neighborhood $\mathcal{X}_\delta \supset\supset \mathcal{X}$ defined in Remark 1, which have dense interior. Using the conclusion of the previous paragraph with $\mathcal{X}$ replaced by $\mathcal{X}_\delta$ gives that for all $x \in \mathcal{X}_\delta$,

$$\Phi^\delta(x) - \Phi^\delta(x^*) \le \left(\frac{1}{\mu_x - \delta} + \frac{L}{2\ell}\right) \cdot \|\nabla\Phi_{1/2\ell}^\delta(x)\| + \frac{\varepsilon_x + \delta}{\mu_x - \delta},$$

where $\Phi^\delta : \mathcal{X}_\delta \to \mathbb{R}$ represents the best-response function $\Phi$ defined with respect to the domain $\mathcal{X}_\delta$. Taking $\delta \downarrow 0$ and using continuity of $\Phi^\delta, \nabla\Phi_{1/2\ell}^\delta$ in $\mathcal{X}_\delta$ as well as continuity of $\nabla\Phi_{1/2\ell}^\delta(\cdot)$ with respect to $\delta$ ensures that (43) holds for any $x \in \mathcal{X}$. $\qquad\square$

**Lemma 13.** For the iterates of two-timescale SGDA, we have

$$\sum_{t=0}^{T-1} \mathbb{E}[\Gamma_t] \le \sqrt{\frac{(D_{\mathcal{X}} + D_{\mathcal{Y}})LT}{\eta_y(1 - \lambda\ell)}} + T\sqrt{\frac{L\sqrt{L^2 + \sigma_x^2}\eta_x}{\eta_y(1 - \lambda\ell)}} + T\sqrt{\frac{(L^2 + \sigma_y^2)\eta_y}{2\lambda(1 - \lambda\ell)}}, \qquad (46)$$

where we recall that $\Gamma_t := \|\nabla\psi_{t,\lambda}(y_t)\|$.

**Proof of Lemma 13.** Adding the inequality (33) for $t = 1, 2, \ldots, T$ and using Jensen's inequality, we have

$$\mathbb{E}[\psi_{T,\lambda}(y_T) - \psi_{0,\lambda}(y_0)] \ge \sum_{t=1}^{T} \eta_y\lambda(1/\lambda - \ell)\mathbb{E}[\Gamma_{t-1}^2] - T \cdot \left(\eta_x L\sqrt{L^2 + \sigma_x^2} + \frac{\eta_y^2(L^2 + \sigma_y^2)}{2\lambda}\right)$$

$$\ge \sum_{t=1}^{T} \eta_y\lambda(1/\lambda - \ell)\mathbb{E}[\Gamma_{t-1}]^2 - T \cdot \left(\eta_x L\sqrt{L^2 + \sigma_x^2} + \frac{\eta_y^2(L^2 + \sigma_y^2)}{2\lambda}\right).$$

For $\lambda \in (0, 1/2\ell)$, we have $1/\lambda - \ell \ge \ell$. Noting that $|\mathbb{E}[\psi_{T,\lambda}(y_T) - \psi_{0,\lambda}(y_0)]| \le (D_{\mathcal{X}} + D_{\mathcal{Y}}) \cdot L$ since $f : \mathcal{X} \times \mathcal{Y} \to \mathbb{R}$ is $L$-Lipschitz, it follows that

$$\sqrt{\sum_{t=0}^{T-1} \mathbb{E}[\Gamma_t]^2} \le \sqrt{\frac{(D_{\mathcal{X}} + D_{\mathcal{Y}})L + T \cdot \left(\eta_x L\sqrt{L^2 + \sigma_x^2} + \eta_y^2(L^2 + \sigma_y^2)/(2\lambda)\right)}{\eta_y(1 - \lambda\ell)}}.$$

The conclusion (46) follows by Cauchy-Schwarz and the inequality $\sqrt{x + y} \le \sqrt{x} + \sqrt{y}$ for $x, y \ge 0$. $\qquad\square$

# D Proofs from Section 5.1

Below we prove [Proposition 2](#). Recall that $V(x,y) = \frac{\langle x, Ry \rangle}{\langle x, Sy \rangle}$ with $R, S$ given by [(12)](#), and for $z = (x,y)$, $F(z) = (\nabla_x V(x,y) - \nabla_y V(x,y))$.

**Proof of [Proposition 2](#).** We first verify that $z^\star$ is the unique Nash equilibrium. Note that

$$\Phi(x) = \max_y V(x,y) = \max \left\{ \frac{-x_1 - \epsilon x_2}{sx_1 + x_2}, \frac{\epsilon x_1}{sx_1 + x_2} \right\} = \frac{\epsilon x_1}{sx_1 + x_2} > 0 \quad \text{for } x_1 > 0$$

$$\Psi(y) = \min_x V(x,y) = \min \left\{ \frac{-y_1 + \epsilon y_2}{sy_1 + sy_2}, \frac{-\epsilon y_1}{y_1 + y_2} \right\} < 0 \quad \text{for } y_1 > 0.$$

The unique global minimum of $\Phi(\cdot)$ over $\mathcal{X} = \Delta(\mathcal{A})$ is at $(x_1, x_2) = (0,1)$, and the unique global maximum of $\Psi(\cdot)$ over $\mathcal{Y} = \Delta(\mathcal{B})$ is at $(y_1, y_2) = (0,1)$. This verifies that $z^\star$ is the unique global Nash equilibrium. The value of the game is $V(x^\star, y^\star) = 0$.

Now consider the point $z = (x,y)$, where $x = y = (1,0)$. Then

$$\langle F(z), z - z^\star \rangle$$

$$= \frac{1}{(x^\top Sy)^2} \cdot \left[ \langle (x^\top Sy) \cdot Ry - (x^\top Ry) \cdot Sy, x - x^\star \rangle + \langle -(x^\top Sy) \cdot (R^\top x) + (x^\top Ry) \cdot (S^\top x), y - y^\star \rangle \right]$$

$$= \frac{1}{s^2} \cdot \left[ -(x^\top Sy)((x^\star)^\top Ry) + (x^\top Ry) \cdot ((x^\star)^\top Sy) + (x^\top Sy) \cdot (x^\top Ry^\star) - (x^\top Ry) \cdot (x^\top Sy^\star) \right]$$

$$= \frac{1}{s^2} \cdot \left[ (x^\top Ry) \cdot ((x^\star)^\top Sy - x^\top Sy^\star) + (x^\top Sy) \cdot (x^\top Ry^\star - (x^\star)^\top Ry) \right]$$

$$= \frac{1}{s^2} \cdot \left[ -1 \cdot (1 - s) + s \cdot (\epsilon - (-\epsilon)) \right]$$

$$= \frac{1}{s^2} \cdot (s + 2\epsilon s - 1),$$

which is negative for sufficiently small $\epsilon$ (in particular, for $\epsilon < \frac{1-s}{2s}$).

Finally, we check that the MVI property

$$\langle F(z), z - \hat{z} \rangle \geq 0 \quad \forall z \in \mathcal{Z} \tag{47}$$

fails for all $\hat{z} = (\hat{x}, \hat{y})$ which are not a Nash equilibrium. For any $\hat{z}$ which is not a Nash equilibrium, either the min-player or max-player can deviate from their policy in a way that increases their utility; we assume without loss it is the min-player (the case for the max-player is symmetric). In particular, there is some $x \in \mathcal{X}$ so that

$$V(x, \hat{y}) = \frac{x^\top R\hat{y}}{x^\top S\hat{y}} < \frac{\hat{x}^\top R\hat{y}}{\hat{x}^\top S\hat{y}} = V(\hat{x}, \hat{y}).$$

It follows that

$$0 > \langle x, (\hat{x}^\top S\hat{y}) \cdot R\hat{y} - (\hat{x}^\top R\hat{y}) \cdot S\hat{y} \rangle$$

$$= \langle x - \hat{x}, (\hat{x}^\top S\hat{y}) \cdot R\hat{y} - (\hat{x}^\top R\hat{y}) \cdot S\hat{y} \rangle$$

$$= (\hat{x}^\top S\hat{y})^2 \cdot \langle x - \hat{x}, \nabla_x V(\hat{x}, \hat{y}) \rangle.$$

It follows that $\langle x - \hat{x}, \nabla_x V(\hat{x}, \hat{y}) \rangle < 0$. For $\alpha \in [0,1]$, define $x_\alpha = (1 - \alpha)\hat{x} + \alpha x$. By continuity of the function $\alpha \mapsto \nabla_x V(x_\alpha, \hat{y})$, there must be some $\alpha \in (0,1)$ so that

$$\frac{1}{\alpha} \langle x_\alpha - \hat{x}, \nabla_x V(x_\alpha, \hat{y}) \rangle = \langle x - \hat{x}, \nabla_x V(x_\alpha, \hat{y}) \rangle < 0.$$

Letting $z := (x_\alpha, \hat{y})$, we obtain that $\langle z - \hat{z}, F(z) \rangle < 0$, violating [(47)](#). $\square$

We remark that an alternative way to verify that $(x^\star, y^\star)$ is a Nash equilibrium in the above proof is as follows: we may calculate that

$$\nabla_x V(x^\star, y^\star) = Ry^\star = \begin{pmatrix} \epsilon \\ 0 \end{pmatrix},$$

$$\nabla_y V(x^\star, y^\star) = R^\top x^\star = \begin{pmatrix} -\epsilon \\ 0 \end{pmatrix},$$

which shows that $x^\star$ satisfies the first-order optimality conditions for minimizing $x \mapsto V(x, y^\star)$, and $y^\star$ satisfies the first-order optimality conditions for maximizing $y \mapsto V(x^\star, y)$. It is then straightforward to check that in fact $x^\star$ is a global minimizer of $x \mapsto V(x, y^\star)$, and $y^\star$ is a global minimizer of $y \mapsto V(x^\star, y)$.

## D.1 Experimental Details

Figure 1(a) and Figure 1(b) use the following game, which is the game from Proposition 2 with $\epsilon = 0.1, s = 0.3$:

$$R = \begin{pmatrix} -1.0 & 0.1 \\ -0.1 & 0.0 \end{pmatrix}, \qquad S = \begin{pmatrix} 0.3 & 0.3 \\ 1.0 & 1.0 \end{pmatrix}.$$

Figure 1(c) and Figure 1(d) use the following game, which is a rounded version of a game we found via a random search:

$$R = \begin{pmatrix} -0.6 & -0.3 \\ 0.6 & -0.3 \end{pmatrix}, \qquad S = \begin{pmatrix} 0.9 & 0.5 \\ 0.8 & 0.4 \end{pmatrix}.$$