[Reviews · NeurIPS 2020]

Review 1

Summary and Contributions: This paper presents the analysis of policy gradient descent algorithms for zero-sum stochastic games. A key feature of this algorithm is that the algorithm is decoupled (i.e. independent for each player). Main theorem (Theorem 1) claims that when learning rates of both players are small and satisfy some pre-specified relations, the algorithm will converges to Nash solution in polynomial episodes. Despite a few aspects that can be further improved, the overall result and contribution look good. See strengths and weakness sections.

Strengths: The paper provides the first analysis of policy gradient descent (both players do PG independently) in the setting of zero-sum stochastic games. The discovery of two-sided gradient dominance condition (Lemma 1) looks interesting, and the use of that to establish convergence is new.

Weaknesses: I am not convinced by the main motivation of this paper for decoupled or independent learning. Specifically, from the communication perspective, once agents can also communicate the actions each other took per round, then each agent can also simulate any coupled algorithm locally (or only coupled online algorithm if has storage limitation). Since agents have to communicate with the oracle or environment in each round anyway, I don't see in practice why communicate the actions in the learning process is that problematic. Second, this paper says that the independent learning is important because it allows the algorithm "being versatile, being applicable even in uncertain environments where the type of interaction and number of other agents are not known to the agent. " I feel this description does not fit the algorithm studied in this paper, thus a bit misleading. The hyperparameters of two agents must be set in a coordinated dependent way. The learning strategy of each agent will only work if assuming the other agent is also doing PG as coordinated. The technical result of this paper is also a little bit weak comparing to several recent results with coupled/dependent algorithms, e.g. [5] and "Learning Zero-Sum Simultaneous-Move Markov Games Using Function Approximation and Correlated Equilibrium". These two prior works do not need to assume minimax mismatch coefficient (Eq(5)), and directly address the exploration problem. Both papers also have relatively sharp dependency on S, A, B, epsilon comparing to the result in this paper which is polynomial in everything. Recent paper "Optimistic Policy Optimization with Bandit Feedback" already addressed both issues in the setting of single-agent MDP. This might be fine for the first paper for PG in stochastic games and independent setting. But it would also be helpful if the authors can cite these related papers, and comment out these technical points other than just saying they are coordinated thus not relevant in the current related work.

Correctness: Yes

Clarity: Yes

Relation to Prior Work: Miss some related work, see comments in the weakness section.

Reproducibility: Yes

Additional Feedback: What is the main difference in proof machinery between this paper and [74]? is there a typo in line 212-213?


Review 2

Summary and Contributions: The authors of this paper study the problem of policy optimization in episodic two-player zero-sum stochastic games (or Markov games) with the underlying model is unknown. This problem is an interesting and important problem in the field of RL. The authors study this problem from the plain lense of optimization and propose a method to simultaneously optimize the policy of maximizing and minimizing players. The proposed approach suggests that each player follow plain gradient descent, but with different learning rates. One with much much smaller learning rate than another one. This way, the slower player follow instant optimum the other player. The proposed approach, along with the analyses, are heavily built on the recent work of [38] and [2]. [38] proposes the two-time scale gradient-based approach for a class of two-player optimization, and [2] show the gradient dominance in MDPs. The author put together the ideas in these works to directly derive theirs. From this sense, the contribution of this paper is minor, but it does not mean it should not be accepted. Of course, the results sound straightforward given [38], and [2], but the authors put the effort, and nicely put these results together, and with their further development provided the final results.

Strengths: The strength of the work is that it is basically proposed the first algorithm (at least to my knowledge) for policy optimization in zero-sum Markov games. The algorithm is simple despite suffering from some slowness (small learning rates). The example in proposition2 is interesting (while the fact was known I guess, sorry I do not have any paper on that on the top of my head)

Weaknesses: There are multiple limitations to this study. 1) is the novelty. Given prior work, the technical contribution of this work is a bit minor, but again this should not be an obstacle to accept this paper. The paper is strong, and I enjoyed reading it. 2) The algorithm is a bit slow, as a two-time scale analysis imposes this slowness. This might be seen as a problem with the two-time scale approach [38] anyway. 3) I wish the authors have provided the orders in their bounds, instead of referring to them in the poly term. I would encourage them to add this. I hope they add it. 4) The whole paper I was waiting for the function approximation part, but I read at the end of the main text that the authors left it for future work. 5) I might understand the authors that eight pages are not enough, but they could have played a better role in explaining and managing the flow in the paper.

Correctness: I did not check the correctness of the claims since the main body of the theoretical results is borrowed from [2] and [38] along with a few others, and I assume the authors carefully deployed these results.

Clarity: Fairly well written. It read well, not much objection. The authors could abstract out the flow and contribution better, but it is alright,

Relation to Prior Work: yes, fairly.

Reproducibility: Yes

Additional Feedback: My evaluation of this paper is not lack anything. The contribution of this paper is important; the problem set up is important. And I would like to see this paper accepted. But since the main body of the contribution comes from the prior work, I was expecting the author to have the complete study of this work, including linear models and function approximation setting. I can totally imagine those extensions could be hard. For that reason, I vote for acceptance, but I might not be able to say the paper is among the top 10%.


Review 3

Summary and Contributions: This paper suggests and analyzes a method to solve zero-sum reinforcement learning games using independent policy gradients algorithms for each of the players. The authors prove that it is possible to reach Nash equilibrium by using learning rates on two different time scales for both players. This follows the reasoning of previous works which show Nash equilibrium guarantees in the case where one player performs policy iteration, where the other computes the best response at each iteration. REBUTTAL: I have read your comments and still believe this work is marginal. My main issue is that the proposed approach is a bit impractical, and the method is not truly independent as suggested in the paper. Also, the analysis is not very tight. However, it is a solid work, on an important problem for which the results to this date are scarce. As such, this is a somewhat novel contribution that can help further researcher in the field. My score remains the same.

Strengths: This paper extends results in zero-sum games to the non-convex non-concave case of multi agent reinforcement learning competitive games. Understanding convergence of MARL algorithm in competitive games can help in wide variety of tasks, especially in the independent decentralized scenario presented in this paper. The result here is somewhat novel as previous works on zero-sum RL games focused on different scenarios such as of linear-quadratic games or as described above - where one player performs policy iteration and the other computes the best response. Policy gradient technique are very common in RL, and extending them further to more complex scenario such as zero-sum games can be beneficial, especially from a practical point of view.

Weaknesses: The main issue that mitigates my enthusiasm is the asymmetry in the players' algorithms which also leads to an asymmetrical Nash equality convergence. The theoretical result is indeed interesting, yet it does not seem to serve a practical algorithm, as the two players need to agree upon learning rates, in contrast to the independence requirement. Moreover, such agreement is not very natural, especially when one player benefits more from this choice. Another caveat of this approach is the dependence on concentrability coefficients - indeed this is a common assumption in the policy gradient literature. Yet, recently, several works proposed to omit this assumption and apply online convex optimization with optimism to solve the policy optimization question. Finally, the number of episodes required for reaching the approximate Nash equilibrium is epsilon^{-12.5}, which is unreasonably slow, which makes me question the practicality of this approach.

Correctness: The results seem reasonable, but I haven't went through all proofs details.

Clarity: The paper is well written. However, I believe the contributions of the paper are not highlighted enough in the exposition. Moreover, the I believe the discussion on last-iterate convergence should be further explained in light of the main theorem of the paper.

Relation to Prior Work: Yes.

Reproducibility: Yes

Additional Feedback: First, I liked reading this work. This is indeed an important and interesting topic! I have several questions regarding this work: 1) One issue is that the dependence in epsilon and in the action-space A and B do not seem very natural to me. Do you have any comments regarding this dependence or lower bounds that can justify this dependence? Is it real, or an artifact of the analysis? 2) Can you further explain why a two time-scale approach is necessary? As you also discuss in your paper, it requires the opponents to cooperate on this matter - any ideas how to omit this requirement? 3) Regarding the concentrabiilty coefficient assumption. Recently, several works shown that this assumption can be reduced by using UCB based optimistic evaluation in policy gradient methods. I wonder whether this be "added" to the analysis for better results.

[Author Response · NeurIPS 2020]

We thank all reviewers for their detailed feedback. We will be sure to address all questions and incorporate all
suggestions in the final version of the paper. Please see individual responses below.
**Reviewer 1:** **1. "I am not convinced by the main motivation of this paper for decoupled or independent**
**learning...".** Motivation for the strongly independent setting comes from a game theoretic perspective. We view
each agent as a strategic party who makes a policy update to improve their own utility each epoch, rather than as an
algorithm with the goal of solving a particular optimization problem. Developing such independent learning protocols
is understood as an interesting and difficult problem in MARL (see Sections 3.2 & 3.4 of [73], as well as the open
problems in Section 6 of [73] about policy gradient methods). The analogous question in the special case of matrix
games has attracted much attention in a line of work starting from [16]. Moreover, our two-timescale approach does not
require coordination: if we think of there being two types agents who play against each other, one of whom takes larger
step sizes than the other, then our result shows that they will converge to a Nash equilibrium. If the agents are to use
vanilla gradient updates each epoch, the two-timescale approach is also necessary: even in the special case of matrix
games, if the agents use equal step sizes, then their policies will not converge (see item 3 for Reviewer 3).
**2. "The technical result of this paper is also a little bit weak comparing to several recent results with coupled**
**algorithms...".** We will cite these works and compare to their quantitative bounds in the final version, but we emphasize
that it is not clear whether any of these results can extend to the independent setting we consider. We view the question
of incorporating optimism (i.e., via exploration bonuses) into the independent learning setting as an interesting direction
for future work. However, we believe that providing guarantees for the vanilla policy gradient method—even though
they depend on the mismatch coefficient—is quite valuable, given that this technique is widely used in practice.
**3. "What is the main difference in proof machinery between this paper and [74]?..."** There is not a typo on line
212. In the different setting of linear-quadratic games, [74] assumes that for each update step the max-player plays a
best response to the min-player's policy, which is updated using gradient descent. We believe that this strategy also
works for the setting we consider, but our main result (Theorem 1) requires a significantly more involved proof: Rather
than using an exact best response, we have to show that if the max-player only performs a single gradient ascent update
at each iteration, its policy stays "sufficiently close" to the best response to the min player's policy.
**Reviewer 2:** **1. Regarding the relation between our work and [2] and [38]:** The analysis for the two-timescale
update in [38] relies on the assumption that $f(x, y)$ is concave with respect to the max-player's iterate $y$. Since concavity
does not hold in stochastic games, our proof requires some new ideas: whereas [38] explicitly considers the movement
over time of the points $y^\star(x_t) := \arg\max_y f(x_t, y)$ and uses the quantity $f(x_t, y^\star(x_t)) - f(x_t, y_t)$ as a potential
function to control the sub-optimality of the iterates $y_t$, we do do not consider the points $y^\star(x_t)$ and must instead use
the gradients of the Moreau envelope of the function $y \mapsto f(x_t, y)$ to play the role of a potential function (Lemmas 9
& 13). Our proof requires the addition of a few other technical contributions not present in [38], including a PL-type
inequality for the max function $x \mapsto \max_y f(x, y)$ (Lemma 12).
**2. "I wish the authors have provided the orders..."** – We provide the explicit bound in Theorem 1a in Section B.2.
**3. On the works which apply MD with optimism to omit concentrability** – Please see item 2 for Reviewer 1.
**4. "I was expecting the author to have the complete study..."** – While we believe that our results stand on their own,
the extension to linear models and beyond is fairly straightforward, and we plan to pursue this in future work.
**Reviewer 3:** **1. On the practicality of our approach and asymmetry:** The problem of proving convergence
guarantees for independent policy gradient algorithms is an important theoretical problem, and our results—while
asymmetric in nature—constitute the first polynomial-time convergence guarantees. Even proving polynomial-time
convergence is a challenging problem that requires new ideas beyond previous work (see item 1 for Reviewer 2).
Though the runtime of our algorithm may preclude immediate practical implications, we believe that our result can
serve as a baseline for future work developing other algorithms, such as extragradient-based ones as we discuss in
Section 5.
**2. "One issue is that the dependence in $\epsilon$ and in the action-space $A$ and $B$ do not seem very natural..."** The likely
sub-optimal dependence on $A, B, \epsilon$ is an artifact of the two-timescale updates, which require that $\eta_x \ll \eta_y$. The fact
that $\eta_x$ is so small slows down convergence: even in the nonconvex-concave case ([38]), in the stochastic setting the
convergence rate is $O(\epsilon^{-8})$; our rate $O(\epsilon^{-12.5})$ is slower because of additional complexities due to the non-concavity
of $y \mapsto f(x, y)$ and the $\varepsilon$-greedy exploration. Again, our aim is to provide the first polynomial time guarantees, and we
hope our work will serve as a baseline for future results developing tighter dependence on problem parameters.
**3. "Can you further explain why a two time-scale approach is necessary?..."** As we discuss at the end of Section 3,
even in the setting of zero-sum matrix games (which is the special case of stochastic games with one state and stopping
probability $\zeta = 1$), it is known that if both players use GDA with equal step sizes, then the dynamics may cycle and
the quantity in (6) can fail to converge to 0 as $N \to \infty$. We believe that one way around two-timescale updates is
for the players to instead use the extragradient update, as discussed in Section 5.1; though equation (EG) allows for
different step sizes $\eta_x, \eta_y$, typically one sets $\eta_x = \eta_y$ for extragradient. We believe that proving convergence rates for
extragradient (with $\eta_x = \eta_y$) in stochastic games is a difficult yet promising direction for future work.
**4. "Regarding the concentrabilty coefficient assumption..."** – please see item 2 for Reviewer 1.

[Meta-Review · NeurIPS 2020]

The reviewers agreed that this is a solid work, on an important problem for which existing results are scarce. However, there were several concerns: - The authors create some confusion in describing their method as "independent" - the agents have to coordinate the learning rates ahead of time. - The analysis is not very tight. I believe that these concerns actually open the door for interesting followup work, and therefore recommend acceptance. I ask the authors to tone down the independence claims in the final version, given the concern above.